# An extra-uterine system to physiologically support the extreme premature lamb

Emily A. Partridge[1,*], Marcus G. Davey[1,*], Matthew A. Hornick[1], Patrick E. McGovern[1], Ali Y. Mejaddam[1], Jesse D. Vrecenak[1], Carmen Mesas-Burgos[1], Aliza Olive[1], Robert C. Caskey[1], Theodore R. Weiland[1], Jiancheng Han[1], Alexander J. Schupper[1], James T. Connelly[1], Kevin C. Dysart[2], Jack Rychik[3], Holly L. Hedrick[1], William H. Peranteau[1] & Alan W. Flake[1]

In the developed world, extreme prematurity is the leading cause of neonatal mortality and morbidity due to a combination of organ immaturity and iatrogenic injury. Until now, efforts to extend gestation using extracorporeal systems have achieved limited success. Here we report the development of a system that incorporates a pumpless oxygenator circuit connected to the fetus of a lamb via an umbilical cord interface that is maintained within a closed 'amniotic fluid' circuit that closely reproduces the environment of the womb. We show that fetal lambs that are developmentally equivalent to the extreme premature human infant can be physiologically supported in this extra-uterine device for up to 4 weeks. Lambs on support maintain stable haemodynamics, have normal blood gas and oxygenation parameters and maintain patency of the fetal circulation. With appropriate nutritional support, lambs on the system demonstrate normal somatic growth, lung maturation and brain growth and myelination.

[1] Center for Fetal Research, Department of Surgery, The Children's Hospital of Philadelphia Research Institute, Room 1116B, 3615 Civic Center Boulevard, Philadelphia, Pennsylvania 19104, USA. [2] Division of Neonatology, Department of Pediatrics, The Children's Hospital of Philadelphia, 3401 Civic Center Boulevard, Philadelphia, Pennsylvania 19104, USA. [3] Division of Cardiology, Department of Pediatrics, The Children's Hospital of Philadelphia, 3401 Civic Center Boulevard, Philadelphia, Pennsylvania 19104, USA. * These authors contributed equally to this work. Correspondence and requests for materials should be addressed to A.W.F. (email: flake@email.chop.edu).

In the United States, extreme prematurity is the leading cause of infant morbidity and mortality, with over one-third of all infant deaths and one-half of cerebral palsy attributed to prematurity[1–3]. Advances in neonatal intensive care have improved survival and pushed the limits of viability to 22 to 23 weeks of gestation. However, survival has been achieved with high associated rates of chronic lung disease and other complications of organ immaturity, particularly in infants born before 28 weeks[1,3]. In fact, with earlier limits of viability, there are actually more total patients with severe complications of prematurity than there were a decade ago[4]. Respiratory failure represents the most common and challenging problem, as gas exchange in critically preterm neonates is impaired by structural and functional immaturity of the lungs. This condition, known as bronchopulmonary dysplasia, is now understood to be related to an arrest in lung development secondary to premature transition from liquid to gas ventilation, explaining why even minimally invasive modes of neonatal ventilation have not reduced the incidence of bronchopulmonary dysplasia[5]. There is clearly an urgent need for a more physiologic approach to support the extreme premature infant.

The concept of extracorporeal support of the fetus is appealing due to the analogy with innate fetal physiology, in which extracorporeal gas exchange is maintained by the placenta. However, the development of an 'artificial placenta' has been the subject of investigation for over 50 years[6,7], with only limited success. The primary obstacles have been progressive circulatory failure due to preload or afterload imbalance imposed on the fetal heart by oxygenator resistance and pump-supported circuits, the use of open fluid incubators resulting in contamination and fetal sepsis and problems related to umbilical vascular access resulting in vascular spasm[6–15]. To address these obstacles we have designed a system consisting of three main components, specifically, a pumpless arteriovenous circuit, a closed fluid environment with continuous fluid exchange and a new technique of umbilical vascular access. Here we demonstrate that extreme premature fetal lambs can be consistently supported in an extracorporeal device for up to 4 weeks without apparent physiologic derangement or organ failure. These results are superior to all previous attempts at extracorporeal support of the extreme premature fetus in both duration and physiologic well-being[16,17].

## Results

### Pilot studies.
A series of pilot studies leading to our final device were performed that identified potential obstacles and allowed sequential design modifications. All pilot studies utilized the pumpless arterial–venous (AV) circuit described below. The primary design modifications were related to the fluid environment and our approach to vascular access. A consistent observation throughout our pilot studies was the haemodynamic stability and efficient gas exchange achieved by the pumpless AV circuit over a wide range of circuit flows. The initial series of experiments were performed in late gestational lambs (125–140 days of gestation; term = 145 days) and utilized an open fluid bath with continuous recirculation of an electrolyte solution (designed to mimic amniotic fluid) through micropore filters. To avoid the potential for umbilical venous spasm we utilized the carotid artery (CA) and jugular vein (JV) for vascular access (Table 1, open CA/JV). These studies lasted 23 to 108 h but were limited by sepsis and cannula-related complications. This led to our second design that included a semi-closed fluid bath with continuous exchange of amniotic fluid, rather than recirculation (Table 1, semi-closed CA/JV). With the improved incubator, five experimental animals with CA/JV cannulation (ranging in age from 120 to 125 days of gestation) were maintained on the system

for 346.6 ± 93.5 h, a marked improvement over the original design. Importantly, one animal was maintained on the circuit for 288 h (120–132 days of gestation) and was successfully weaned to spontaneous respiration, with long-term survival confirming that animals can be transitioned to normal postnatal life after prolonged extra-uterine support. However, sepsis remained limiting in 3 of 5 lambs resulting in design of a closed fluid circuit (the Biobag). With introduction of the Biobag, pilot studies were performed with the aim to apply our system to earlier gestational fetuses. From the perspective of lung development, lambs at 100 − 115 days of gestation are in the mid to late canalicular phase of lung development[18], which is the biological equivalent of the 22–24-week gestation premature human infant[19]. However, in 110-day gestational age (GA) lambs with CA/JV cannulation, diminishing circuit flows and progressive oedema developed within the first few days. Low circuit flows were due to reduced perfusion pressures across the oxygenator owing to lower mean arterial pressure in earlier GA lambs combined with elevated right-sided venous pressures in the 110-day GA CA/JV animals relative to published *in utero* controls (inferior vena cava pressures 9.6 ± 2 mm Hg versus 4 ± 2 mm Hg, respectively)[20–22]. To offload the right atrium, we opted to utilize the umbilical vein (UV) for venous inflow to mimic normal fetal umbilical venous return and improve streaming of oxygenated blood across the foramen ovale[20,21]. To avoid umbilical venous spasm[23–25], we advanced the umbilical venous cannula to a position with the tip just inside the abdominal fascia. CA/UV cannulation resulted in the stable support of five 106–113-day GA lambs for 13 to 26 days in the Biobag (Table 1, Biobag, CA/UV). All five lambs demonstrated long-term haemodynamic stability and stable circuit flows and oxygenation parameters (CA/UV group in Fig. 2). However, flow to the oxygenator in CA/UV lambs was well below the normal physiologic flow to the placenta (70–120 versus 150–200 ml kg$^{-1}$ min$^{-1}$)[26–28], primarily due to the inherently small-caliber carotid artery. The limiting problem of carotid vascular inflow led to the development of our final device to achieve physiologic fetal support (Fig. 1 and Supplementary Movie 1). The three components of the system are described below.

### A pumpless arteriovenous circuit.
From the inception of the study, we reasoned that a pumpless circuit—in which blood flow is driven exclusively by the fetal heart—combined with a very low resistance oxygenator would most closely mimic the normal fetal/placental circulation. In most of our studies we utilized a small-volume, near-zero-resistance oxygenator and short segments of circuit tubing to minimize surface area and priming volumes. This system is comparable to the volume of the placenta itself—the reported placental blood volume of the sheep is 23.1 to 48.1 ml per kg[29], and most of the studies in this report utilized the Quadrox-ID Pediatric oxygenator (Maquet Quadrox-ID Pediatric Oxygenator: Maquet Cardiopulmonary AG, Rastatt, Germany) that has a priming volume of 81 ml. Recently, with smaller lambs (0.5 to 1 kg) we have utilized a modified Quadrox Neonatal oxygenator (Maquet Quadrox-I Neonatal and Pediatric Oxygenator: Maquet Cardiopulmonary AG) with a priming volume of 38 ml. Thus, our circuit priming volumes for 1 to 3 kg lambs are within the normal placental blood volume range. Throughout the development of our device, with the exception of our earlier gestational CA/JV animals described above, all animals demonstrated complete haemodynamic stability, without need for vasopressors or evidence of progressive acidosis or circulatory failure.

### A closed sterile fluid environment.
To further address issues of sterility, size adaptability and efficiencies of space and fluid volume, a 'Biobag' design was developed—a single-use, completely closed

**Table 1 | Overview of experimental animals.**

| Incubator design/ Cannulation strategy/ Animal # | Cannula size, French (arterial/ venous) | GA (weight, kg) at cannulation and at study end | Length of run (h) | Complications during run | Culture results* | Outcome and pathology† |
|---|---|---|---|---|---|---|
| Open CA/JV 1 | 8/8 | 140 (3.62) N/A | 23 | Bacterial overgrowth in AF; Sepsis | AF +++ Blood − | Did not survive to delivery/ventilation. Diffuse pulmonary inflammation and hemorrhage. |
| Open CA/JV 2 | 8/8 | 135 (4.89) N/A | 71 | Bacterial overgrowth in AF; Sepsis | AF +++ Blood + | Did not survive to delivery/ventilation. Diffuse pulmonary inflammation. |
| Open CA/JV 3 | 12/10 | 135 (3.49) N/A | 96 | Bacterial overgrowth in AF | AF +++ Blood + | Good gas exchange on ventilator. Diffuse pulmonary inflammation. |
| Open CA/JV 4 | 10/12 | 130 (4.24) N/A | 51 | Cardiac arrhythmia | AF − Blood − | Did not survive to delivery/ventilation. Diffuse shower emboli in lungs, heart, liver, bowel.‡ |
| Open CA/JV 5 | 10/12 | 120 (3.20) N/A | 108 | Cannula displacement | AF − Blood − | Did not survive to delivery/ventilation. Normal organ histology. |
| Semi-closed CA/JV 1 | 10/10 | 125 (3.17) 134 (3.9) | 209 | Hypoxia during midazolam Infusion | AF − Blood − | Good gas exchange on ventilator. |
| Semi-closed CA/JV 2 | 10/10 | 120 (3.20) 135 (4.20) | 360 | Bacterial overgrowth in AF | AF +++ Blood − | Good gas exchange on ventilator. Diffuse pulmonary inflammation. |
| Semi-closed CA/JV 3 | 10/12 | 120 (3.30) 136 (4.50) | 372 | Bacterial overgrowth in AF | AF +++ Blood − | Good gas exchange on ventilator. Diffuse pulmonary inflammation. |
| Semi-closed CA/JV 4 | 10/10 | 120 (3.30) 132 (3.85) | 288 | Minor bleeding at cannulation site | AF +++ Blood − | Good gas exchange on ventilator. Extubated. Long-term survivor with normal MRI brain at 1 year. |
| Semi-closed CA/JV 5 | 10/12 | 120 (2.89) 140 (4.20) | 504 | Gastrointestinal bleeding | AF − Blood − | Bowel obstruction preventing extubation. Good gas exchange on ventilator. |
| Biobag CA/UV 1 | 8/10 | 112 (2.03) 133 (2.99) | 504 | Gastrointestinal bleeding | AF + Blood − | Good gas exchange on ventilator. |
| Biobag CA/UV 2 | 8/10 | 113 (1.99) 126 (2.67) | 310 | Oxygenator failure (not heparinized); neuro dysfunction | AF − Blood − | Did not survive to delivery/ventilation. |
| Biobag CA/UV 3 | 8/10 | 113 (1.88) 132 (3.00) | 452 | Oxygenator failure (not heparinized) | AF − Blood − | Did not survive to delivery/ventilation. |
| Biobag CA/UV 4 | 8/10 | 106 (1.00) 132 (2.51) | 618 | Bleeding from cord; Acute kidney injury | AF − Blood − | Good gas exchange on ventilator. |
| Biobag CA/UV 5 | 8/12 | 113 (1.82) 134 (2.72) | 502 | None | AF + Blood − | Good gas exchange on ventilator. |
| Biobag UA/UV 1 | 12(x2)/12 | 106 (2.01) 132 (3.39) | 621 | None | AF + Blood − | Good gas exchange on ventilator. Survived off ventilator for 10 hours. |
| Biobag UA/UV 2 | 12(x2)/12 | 105 (1.50) 133 (2.77) | 669 | Transient RV failure; AF sterility breaches | AF + Blood − | Good gas exchange on ventilator. Mild pulmonary inflammation. |
| Biobag UA/UV 3 | 12(x2)/12 | 110 (1.55) 135 (2.95) | 600 | Biobag entry for bladder catheterization | AF + Blood − | Tetralogy of Fallot with hypoxemia on ventilator. Moderate pulmonary inflammation. |
| Biobag UA/UV 4 | 12(x2)/12 | 116 (2.39) 135 (3.53) | 468 | None | AF + † Blood − | Good gas exchange on ventilator. Severe upper GI bleed on DOL1. Mild pulmonary inflammation. |
| Biobag UA/UV 5 | 12(x2)/12 | 116 (2.31) 144 (4.52) | 678 | None | AF + Blood − | Good gas exchange on ventilator. |
| Biobag UA/UV 6 | 12(x2)/12 | 117 (2.00) 137 (2.65) | 480 | Biobag entry for bladder catheterization | AF + ‡ Blood − | Good gas exchange on ventilator. Moderate pulmonary inflammation. |
| Biobag UA/UV 7 | 12(x2)/12 | 111 (1.80) 139 (3.17) | 672 | None | AF + † Blood − | Hypoxemia at delivery. Pulmonary hypoplasia. |
| Biobag UA/UV 8 | 12(x2)/12 | 106 (1.31) 132 (2.71) | 627 | Bleeding from cord | AF + † Blood − | Hypoxemia on ventilator. |

AF, amniotic fluid; DOL 1, day of life 1; GI, gastrointestinal; MRI, magnetic resonance imaging; NA, not available.
*AF = amniotic fluid; +++, gross contamination; +, scant contamination.
†No growth on AF cultures after daily antibiotic injections into Biobag.
‡Scant growth on AF cultures after daily antibiotic injections into Biobag.

system that minimizes amniotic fluid volumes and can be customized to more closely replicate the size and shape of the uterus. The Biobag consists of polyethylene film that is translucent, sonolucent and flexible to permit monitoring, scanning and manipulation of the fetus as necessary. An open, sealable side was incorporated to facilitate insertion of the fetus at the time of

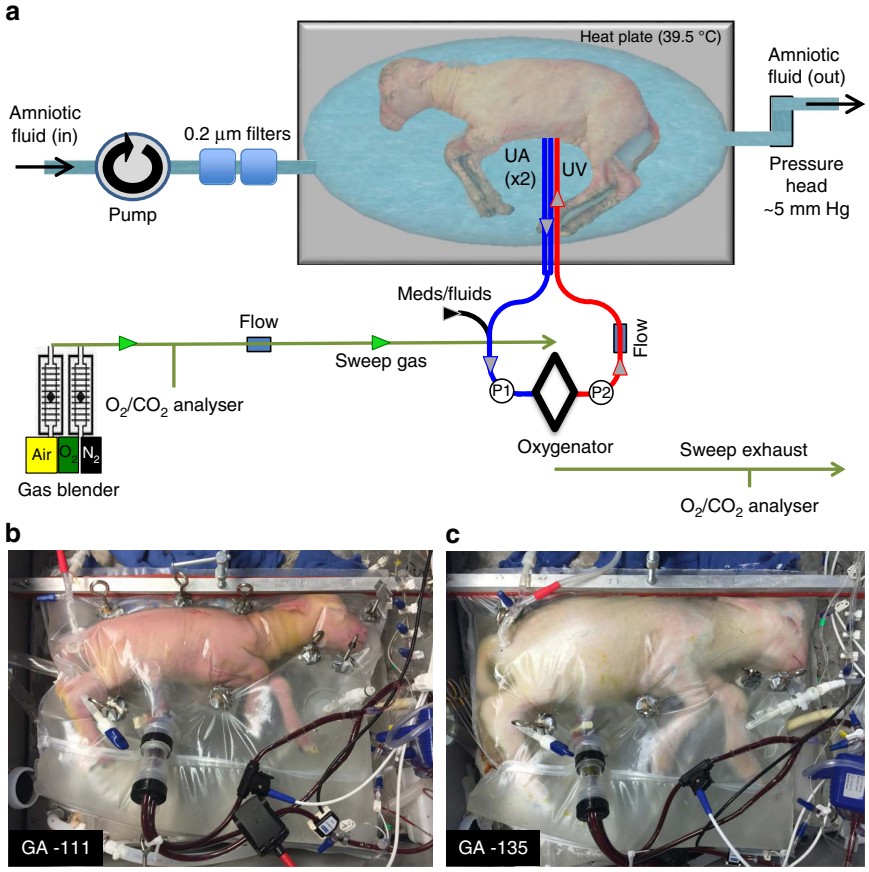

**Figure 1 | UA/UV Biobag system design. (a)** Circuit and system components consisting of a pumpless, low-resistance oxygenator circuit, a closed fluid environment with continuous fluid exchange and an umbilical vascular interface. (**b**) Representative lamb cannulated at 107 days of gestation and on day 4 of support. (**c**) The same lamb on day 28 of support illustrating somatic growth and maturation.

cannulation, and various water-tight ports were designed to accommodate cannulas, temperature probes and sterile suction tubing. After cannulation, the Biobag is sealed and transferred to a mobile support platform that incorporates temperature and pressure regulation, padding and the fluid reservoirs and fluid exchange circuitry. The development of the Biobag essentially solved the problem of gross fluid contamination, and has eliminated pneumonia on lung pathology. Throughout the subsequent experiments, low-level amniotic fluid contamination was observed only in circumstances where Biobag re-entry was required. When this occurred, contamination could be cleared by increasing the fluid exchange rate and injecting antibiotics into the bag fluid on a daily basis.

**Umbilical vascular access**. To more closely approximate flow dynamics *in utero*, carotid cannulation was abandoned in favour of double umbilical artery and single umbilical vein cannulation (abbreviated UA/UV) cannulation. We developed a technique for umbilical cord vessel cannulation that maintains a length of native umbilical cord (5–10 cm) between the cannula tips and the abdominal wall, to minimize decannulation events and the risk of mechanical obstruction (Fig. 1b,c). The umbilical and venous cannulas are only 2 cm long, most of which is used for securing the cannulas, and therefore the interface is functionally end to end. Umbilical cord spasm was mitigated by a combination of topical papaverine administration, atraumatic operative technique and maintaining warmth and physiologic oxygen saturation of the umbilical venous inflow

on initiation of circuit flow. The Biobag was modified to accommodate exclusively umbilical cannulas (Fig. 1b,c and Supplementary Movie 1).

**Physiologic extracorporeal support of the fetus**. The combination of the pumpless oxygenator circuit, the closed fluid circuit and Biobag and umbilical cord access constitute our device. We have run 8 lambs with maintenance of stable levels of circuit flow equivalent to the normal flow to the placenta. We have run 5 fetuses from 105 to 108 days of gestation for 25–28 days, and 3 fetuses from 115 to 120 days of gestation for 20–28 days (Table 1, Biobag UA/UV). The longest runs were terminated at 28 days due to animal protocol limitations rather than any instability, suggesting that support of these early gestational animals could be maintained beyond 4 weeks. Haemodynamic and circuit flow parameters for all eight UA/UV lambs are summarized in Fig. 2 and compared directly with CA/UV lambs. The UA/UV lambs demonstrated levels of circuit flow comparable to what is considered normal placental flow (Fig. 2c, 150–250 ml kg$^{-1}$ min$^{-1}$) throughout the duration of their runs. With physiologic circuit flows in UA/UV animals, we were able to maintain lower post-membrane saturation (Fig. 2d) with normal fetal oxygen delivery (Fig. 2f) and reduced transfusion requirement (Fig. 2e) relative to CA/UV lambs. To prevent excessive oxygen delivery, we typically lowered initial sweep-gas oxygen concentration to 11–14% by blending nitrogen with room air.

Normal oxygen delivery is required for physiologic metabolic support, substrate utilization and growth and development.

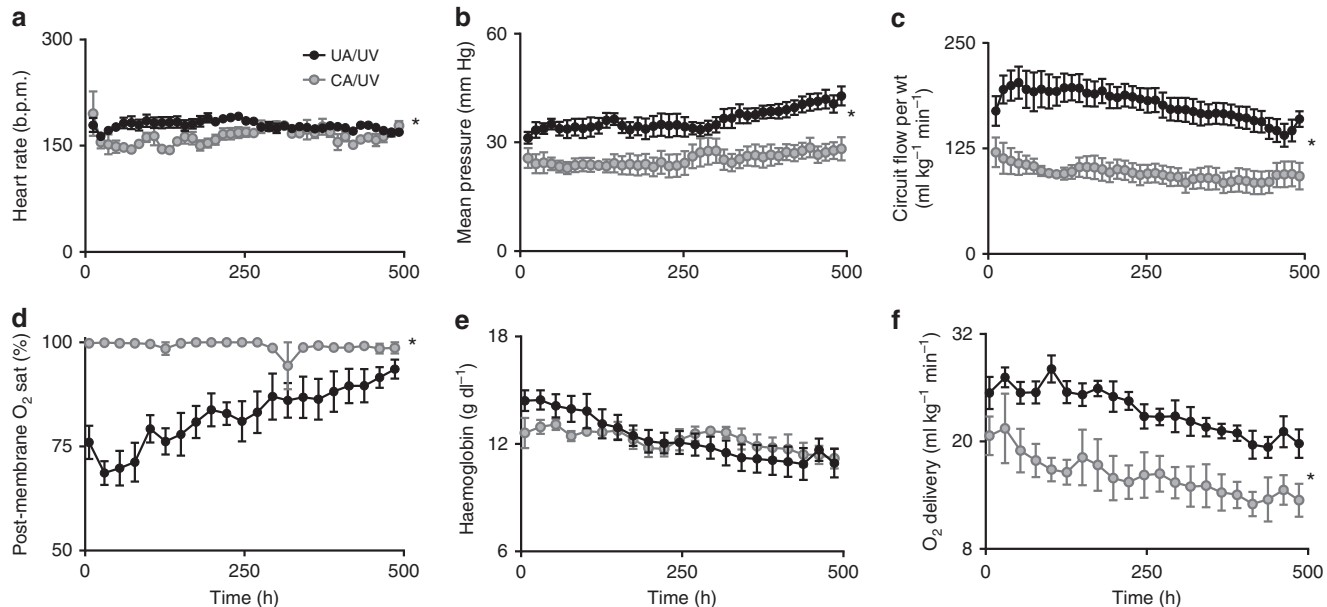

**Figure 2 | Haemodynamic and oxygen parameters in CA/UV lambs versus UA/UV lambs.** CA/UV lambs data represented by grey circles ($n=5$) and UA/UV lambs as black circles ($n=8$). (**a**) Heart rate. (**b**) Mean pre-membrane oxygenator pressure calculated as $1/3$ systolic $+ 2/3$ diastolic. (**c**) Body weight-adjusted circuit flow. (**d**) Post-membrane oxygen saturation (sat). (**e**) Haemoglobin. (**f**) Oxygen delivery. Data are presented as mean ± s.e.m. Statistically significant difference between groups in (**a–f**) is denoted by *$P < 0.05$ (analysis of variance (ANOVA)).

Figure 3a shows the calculated parameters related to oxygen delivery for our UA/UV lambs on both the 5 earlier gestation lambs (105 to 108 days of gestation) and the 3 later gestation lambs (115 to 120 days of gestation). The values for both groups are comparable to the previously published normal values for fetal lambs[30]. During studies, efforts were made to maintain normal fetal oxygen tension and carbon dioxide exchange while providing normal oxygen delivery (Fig. 3b). In CA/UV lambs a progressive fall in haemoglobin (Hgb) levels was noted, ultimately requiring transfusion with a large volume of adult blood ($\sim 40$ ml kg$^{-1}$ per week) to maintain $O_2$ delivery. We reasoned that this was likely due to impaired erythropoietin production from the fetal liver due to supraphysiologic $O_2$ content of the post-membrane UV blood[31]. Subsequent experiments with the first three UA/UV animals demonstrated reduced transfusion requirement ($\sim 10$ ml kg$^{-1}$ per week), likely due to lower $PaO_2$ in UV blood. In the last five UA/UV animals we administered daily erythropoietin that slowed the progression of anaemia and nearly eliminated ($\sim 6$ ml kg$^{-1}$ per week) or, in the last 3 animals, completely eliminated the need for blood transfusion (Fig. 3c). Fetuses had normal pH values (Fig. 3d) and lactate levels (Fig. 3e) throughout the studies.

Daily echocardiography confirmed physiologic cardiac outputs and maintenance of the fetal cardiac circulation throughout the UA/UV runs, with near-normal ductus arteriosus flows (Fig. 4a–c), patency and flow through the ductus venosus and right to left shunting through the foramen ovale (Supplementary Movies 2–4)[32]. Cardiac contractility was excellent and chamber and vena caval size could be used as an indicator of general volume status, allowing adjustments in fluid administration. An experienced fetal cardiologist (J.R.) reviewed all echocardiographic data and agreed that cardiac function appeared grossly normal in all respects.

**Growth and organ maturation.** The nutrition provided via the circuit was based on substrate uptake of late-gestation fetal lambs[33–35], and hence consisted predominantly of carbohydrate and amino acid, with trace lipid. Our strategy in CA/UV and UA/UV lambs was to titrate dextrose and amino acid administration to levels of plasma glucose ($< 40$ mg dl$^{-1}$) and blood urea nitrogen ($< 30$ mg dl$^{-1}$) to avoid an osmotic diuresis and/or hyperosmolar state. Substrate tolerance generally correlated with oxygen delivery, and at relatively higher oxygen delivery, UA/UV lambs tolerated physiologic levels of substrate delivery (Fig. 5a) and demonstrated growth comparable to breed-matched controls (Fig. 5b,c). From a qualitative perspective, there was obvious growth and maturation with prolonged runs. Animals opened their eyes, became more active, had apparently normal breathing and swallowing movements, grew wool and clearly occupied a greater proportion of space within the bags (Fig. 1b,c). The addition of insulin infusions in the last two lambs further improved substrate utilization and allowed administration of higher caloric loads with further enhancement of fetal growth.

As a surrogate for organ maturation in our system, we assessed lung maturation in UA/UV lambs by detailed morphometric analysis (Fig. 6a–e), histologic assessment (Fig. 6f–i), surfactant protein B analysis (Fig. 6j,k) and analysis of function after birth (Fig. 6m). Lambs at $106 - 113$ days of gestation are in the mid to late canalicular phase of lung development[18], which is the biological equivalent of the 23–24-week gestation premature human infant[19]. Morphometric analysis demonstrated progression from the canalicular to saccular stages of lung development in parallel with age-matched normal control lambs (Fig. 6a–e). From a functional perspective, lambs were easily ventilated after removal from the circuit, and nearly comparable to 141-day GA control lambs delivered by caesarean section and immediately ventilated (Fig. 6m). Other metabolic parameters reflective of organ function and nutritional status were surprisingly stable despite the known maternal contributions to hepatic and renal function (Table 2). Bilirubin levels and liver function tests showed only very mild elevation or remained within the normal range.

Brain growth and development were also assessed in UA/UV animals. Brains were grossly normal appearing after runs with no difference in brain-to-body weight ratios in experimental animals versus age-matched *in utero* controls (Fig. 7a). To assess brain maturation we analysed gyral thickness that also demonstrated no difference from controls (Fig. 7b). Biparietal diameter has been reported as a surrogate for brain growth[36] and the growth curve for lambs maintained on our system was similar to the expected curve determined from weight-based calculations (Fig. 7c)[37]. As the effects of exogenous insulin on brain growth are unknown, animals receiving insulin were excluded from brain growth assessments ($n = 2$). On whole-brain sectioning we observed no evidence of haemorrhage or infarct in any of the UA/UV animal brains ($n = 5$). Finally, to assess ischaemic injury and global brain integrity on the surviving lamb, postnatal T1-, T2- and diffusion-weighted magnetic resonance imaging sequences were performed at 6 months of age (Fig. 7d). Despite utilization of the CA in this animal, there was no evidence of ischaemia or structural defects. To confirm these findings at a histologic level we performed routine haematoxylin and eosin staining on critical brain regions (Fig. 8a) and, as white matter is particularly sensitive to ischaemic injury, we assessed critical regions of the brain by densitometry of myelin-stained sections[38–40] in UA/UV animals. There was no difference in myelin density in any of the brains analysed (Fig. 8b–d). In addition, neuropathologists from two separate institutions were unable to identify any histologic lesions associated with ischaemia, infarction or demyelination when blinded to experimental and control tissues.

From a gross functional level, the UA/UV animals demonstrated normal or increased movement, sleep/wake cycles, intermittent breathing and swallowing and generally appeared comfortable and nondistressed. Fetal breathing movements were noted regularly throughout the incubation period. As one measure of neurologic development, we compared ocular electromyography (EMG) in two chronically catheterized fetal lambs[41], with two lambs maintained in the system over the same range of gestation. A developmental progression from fragmented to consolidated sleep/wake cycles between the two gestational ages is apparent in both *in utero* and experimental animals (representative data shown in Fig. 8e). Finally, although brain perfusion was not directly assessed, the middle cerebral artery pulsatility index was routinely measured and correlated with oxygen delivery (Fig. 8f), consistent with normal autoregulation of cerebral blood flow. Taken together with our observation of normal cardiac outputs, oxygen delivery and fetal circulatory pathways, we feel our limited data to this point is encouraging with respect to cerebral perfusion and brain development. It is important to note however that there are important differences between fetal lamb brain maturation and human brain maturation, most important of which is the earlier maturation of the germinal matrix in the lamb (70 days)[42]. Thus, the risk of intracranial haemorrhage cannot be assessed in the ovine model[43]. In addition, long-term neurologic follow-up is difficult in our model, due to difficulties with survival of premature lambs and to the limitations in assessment of lamb neurologic function. Thus, any conclusions regarding neurologic development must be qualified.

## Discussion

A pumpless circuit powered by the fetal heart is not a new concept and has been the initial approach taken by many investigators[23–25,44–46]. The advantages include simplicity, absence of pump-induced haemolysis and the potential for at least some autoregulation of circuit blood flow. The disadvantages of pumpless systems include cardiac failure due to afterload imbalance if the circuit/oxygenator has supraphysiologic resistance, or the potential for high-output cardiac failure if the oxygenator has subphysiologic resistance. Most attempts have been limited by subphysiologic circuit flows and rapid haemodynamic decompensation despite the use of vasopressor support and other measures[24,25,44,46]. Recently, Miura et al.[45] reported survival of lamb fetuses on a pumpless parallelized circuit up to a predetermined limit of 60 h. In this report, the animals remained relatively haemodynamically stable. However, external flow regulators were required to reduce blood flow through the circuit to reverse lactic acidosis, and evidence of white matter brain injury was observed on histology. Most investigators have added pumps to arteriovenous systems to address the limitation of subphysiologic flow in pumpless systems[7–9,11–15]. While most attempts have been short-lived culminating in circulatory failure, prolonged survival of up to 543 h was achieved in two fetal goats[12]. However, to achieve this result, the animals required dialysis, continuous paralysis, were hydropic and ultimately succumbed to respiratory failure. Hollow fibre plate technology has allowed the development of near-zero-resistance oxygenators, enabling our pumpless system. In contrast to previous studies, throughout the development of our system, we saw no instances of cardiac failure in our developmentally relevant lambs (105 days of gestation and beyond). In fact, a low level of resistance developed across the circuit when umbilical vascular access was used, consistent with some autoregulation of blood flow.

A major concern in premature infants is intracranial haemorrhage, raising concern for the use of anticoagulation in extracorporeal support systems. We use substantially reduced heparin doses compared with conventional extracorporeal membrane oxygenation (ECMO) to maintain activated clotting times in the 150–180 s range. We attribute our reduced heparin dosage to the reduced surface area of our circuit and the inclusion of a heparin-bound coating to all of the blood contacting components. In the future there will likely be non-heparin-based coatings that will further improve safety. However, there is also evidence that germinal matrix haemorrhage is related to positive pressure ventilation, inotrope use[47] and other interventions in the extreme premature infant[48]. Thus, physiologic support in a extracorporeal system without ventilation or pressors may, in itself, reduce the likelihood of haemorrhage, making prediction of the impact of our system on intracranial haemorrhage difficult.

A critical feature of our system is the closed fluid environment with continuous fluid exchange. This has many advantages, the most important of which is preservation of fluid-filled lungs and the normal glottic resistance required for maintenance of normal airway pressures and lung growth and development[5,19]. In addition, a fluid environment maintains the protective barrier between the fetus and the outside world. Finally, fetal swallowing of amniotic fluid helps maintain fetal fluid homeostasis and potentially may provide an additional route for nutrition. Although we used a simple electrolyte solution in this study, amniotic fluid contains many trophic factors and other components that may be beneficial to the fetus. The development of an optimal 'amniotic fluid' for use in the device is an important focus of our current research. Continuous fluid exchange is employed to remove waste and maintain sterility and is analogous to the physiologic turnover of amniotic fluid. The rate of fluid exchange can be increased to clear any contamination that occurs and we have not had any clinically significant infections since development of the Biobag. Disadvantages of a fluid environment include limited access to the fetus by caregivers (that is, physical examination, blood draws, haemodynamic monitoring) with contamination of the environment if

such access is required, and lack of rapid access in the case of an emergency. In our system, we can examine the fetus in great detail by ultrasound that is superior to physical examination, and can access the circuit for all our vascular access needs (haemodynamic monitoring, blood draws and fluid/nutritional support). Sterile access ports for suction of meconium and other purposes are included, and we have designed a clinical device that could be rapidly opened should the fetus need to be resuscitated.

| a | Control † | GA 105–111 days | | | | | | GA 115–120 days | | | |
|---|---|---|---|---|---|---|---|---|---|---|---|
| | *In utero* (GA 125 d) | 1 | 2 | 3 | 7 | 8 | Mean ± s.e.m. | 4 | 5 | 6 | Mean ± s.e.m. |
| Gestational age (days) at cannulation | NA | 106 | 105 | 110 | 111 | 106 | 108 ± 1 | 116 | 116 | 117 | 116 ± 0 |
| Duration of run (h) | NA | 621 | 669 | 600 | 672 | 627 | 638 ± 14 | 468 | 678 | 480 | 542 ± 68 |
| Haemoglobin (g dl$^{-1}$) | 9.0 ± 1.0 | 11.4 | 10.4 | 9.8 | 12.7 | 10.8 | 11.0 ± 0.5 | 14.7 | 14.2 | 12.2 | 13.7 ± 0.8 |
| 'Circuit' flow per weight (ml kg$^{-1}$min$^{-1}$) | 233 ± 40 | 156 | 190 | 194 | 179 | 226 | 189 ± 11 | 135 | 142 | 134 | 137 ± 3 |
| Post-membrane $O_2$ saturation (%) | 83.2 ± 5.1 | 79.4 | 88.4 | 87.9 | 72.5 | 77.4 | 81.1 ± 3.1 | 85.6 | 87.7 | 94.7 | 89.3 ± 2.8 |
| Total $O_2$ delivery (ml kg$^{-1}$min$^{-1}$) | 22.3 ± 2.2 | 18.3 | 23.8 | 22.3 | 22.4 | 26.3 | 22.6 ± 1.3 | 23.0 | 23.5 | 21.2 | 22.6 ± 0.7 |
| Total $O_2$ consumption (ml kg$^{-1}$min$^{-1}$) | 7.0 ± 2.7 | 6.5 | 10.0 | 9.4 | 7.6 | 10.2 | 8.7 ± 0.7 | 9.4 | 8.2 | 9.1 | 8.9 ± 0.4 |
| Total $O_2$ extraction (%) | 32.8 ± 7.5 | 34.7 | 42.3 | 42.4 | 34.1 | 40.0 | 38.7 ± 1.8 | 41.5 | 35.1 | 43.5 | 40.0 ± 2.5 |
| Umbilical artery $P_aCO_2$ (mm Hg) | 44.9 ± 2.7 | 44.4 | 39.3 | 42.2 | 40.9 | 39.7 | 41.3 ± 1.8 | 43.1 | 43.2 | 39.5 | 41.9 ± 1.2 |
| Umbilical artery $P_aO_2$ (mm Hg) | 20.7 ± 2.0 | 24.7 | 21.9 | 25.8 | 20.6 | 19.9 | 22.6 ± 1.1 | 20.8 | 21.9 | 22.7 | 21.8 ± 0.6 |
| Umbilical artery $O_2$ saturation (%) | 54.6 ± 4.5 | 54.3 | 51.2 | 50.3 | 48.3 | 46.9 | 50.2 ± 1.3 | 50.2 | 57.6 | 54.2 | 54.0 ± 2.1 |
| Umbilical artery $O_2$ content (ml dl$^{-1}$) | 6.6 ± 0.8 | 8.3 | 7.1 | 6.6 | 8.3 | 6.9 | 7.4 ± 0.4 | 9.9 | 11.0 | 8.9 | 9.9 ± 0.6 |
| Umbilical artery pH | 7.39 ± 0.04 | 7.42 | 7.39 | 7.38 | 7.36 | 7.38 | 7.39 ± 0.01 | 7.35 | 7.39 | 7.37 | 7.37 ± 0.01 |
| Plasma lactate (mmol l$^{-1}$) | 1.8 ± 0.5 | 0.8 | 0.8 | 0.9 | 1.0 | 1.4 | 1.0 ± 0.1 | 1.0 | 1.0 | 0.9 | 1.0 ± 0 |

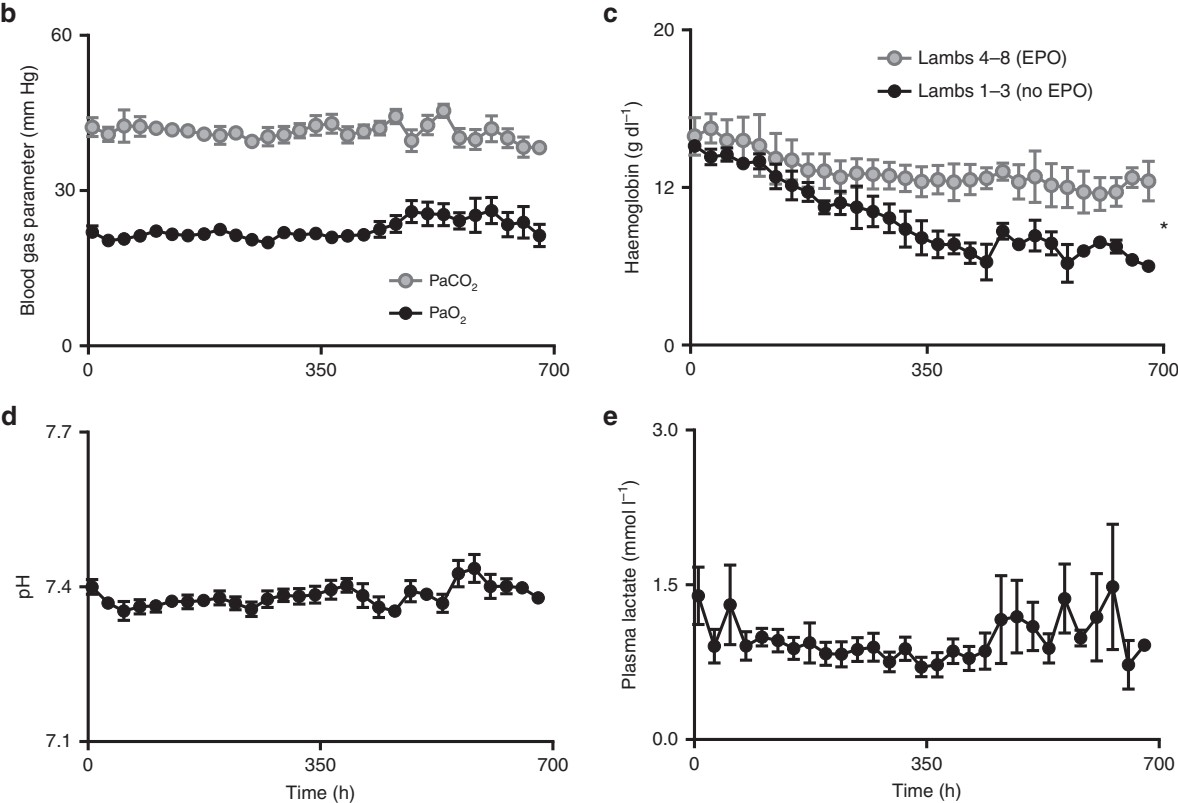

**Figure 3 | Oxygen parameters in UA/UV lambs.** Data from $n = 8$ lambs. (**a**) Haemodynamic and laboratory parameters. (**b**) Pre-membrane $P_aCO_2$ and $P_aO_2$. (**c**) Haemoglobin levels with and without erythropoietin (EPO). (**d**) Pre-membrane pH. (**e**) Plasma lactate. † *In utero* control values in (**a**) derived from measured data in ref. 18. Data in (**b**-**e**) are presented as mean ± s.e.m. Statistically significant difference between groups in (**c**) denoted by *$P < 0.05$ (analysis of variance (ANOVA)).

**a**

| | Control | | | UA/UV | | | | | | | | Mean ±s.e.m. |
|---|---|---|---|---|---|---|---|---|---|---|---|---|
| | In utero (micro-sphere) GA 125d* | In utero (echo) GA 109d | In utero (echo) GA 135d | 1 | 2 | 3† (TOF) | 4 | 5 | 6 | 7 | 8 | |
| CCO (ml kg⁻¹ min⁻¹) | 490 | 562 | 581 | 586 | 666 | 760 | 503 | 467 | 635 | 603 | 699 | 594 ± 32‡ |
| RV output (ml kg⁻¹ min⁻¹) | 321 | 331 | 363 | 339 | 382 | 296 | 307 | 281 | 366 | 329 | 362 | 338 ± 13‡ |
| LV output (ml kg⁻¹ min⁻¹) | 169 | 231 | 218 | 247 | 284 | 463 | 196 | 186 | 268 | 274 | 338 | 256 ± 20‡ |
| RV:LV output ratio | 1.90 | 1.43 | 1.66 | 1.43 | 1.35 | 0.67 | 1.57 | 1.53 | 1.39 | 1.22 | 1.09 | 1.37 ± 0.06‡ |
| DA flow (ml kg⁻¹ min⁻¹) | 289 | 288 | 182 | 90 | 199 | 180 | 239 | 189 | 283 | 178 | 211 | 196 ± 20 |
| DA flow: RV output ratio | 0.90 | 0.87 | 0.50 | 0.25 | 0.53 | 0.62 | 0.79 | 0.70 | 0.76 | 0.55 | 0.60 | 0.60 ± 0.06 |
| Umbilical PₐO₂ (mm Hg) | 20.7 | NA | NA | 24.7 | 21.9 | 25.8 | 20.8 | 21.9 | 22.7 | 20.6 | 19.9 | 22.3 ± 0.7 |

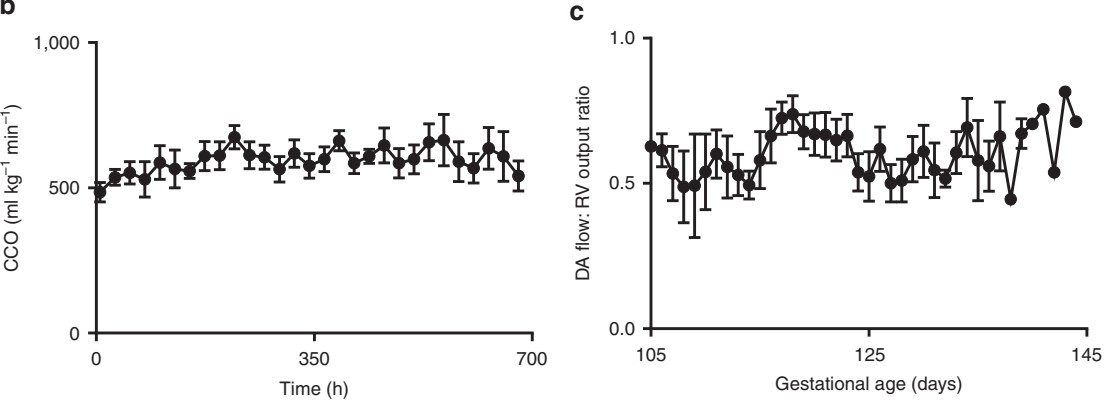

**Figure 4 | Echocardiographic parameters in UA/UV lambs.** Data from $n = 8$ lambs. (**a**) Echocardiographic parameters. (**b**) Combined cardiac output. (**c**) Ductus arteriosus (DA) flow to right ventricular output ratio. Data in (**b**,**c**) are presented as mean ± s.e.m. *In utero* microsphere data in (**a**) derived from measured data in ref. 20. †Lamb 3 noted to have Tetralogy of Fallot (TOF) with restricted pulmonary flow before cannulation was excluded from mean value calculations (‡) due to TOF physiology.

**a**

| | Control | UA/UV no insulin | | | | | | | UA/UV insulin | | |
|---|---|---|---|---|---|---|---|---|---|---|---|
| | In utero† | 1 | 2 | 3 | 4 | 5 | 6 | Mean ±s.e.m. | 7 | 8 | Mean ±s.e.m. |
| Carbohydrate supplied (g kg⁻¹ day⁻¹) | 9 | 10.2 | 13.9 | 13.1 | 13.4 | 10.3 | 10.5 | 13.6 ± 1.3 | 15.9 | 21.3 | 18.6 ± 2.7 |
| Plasma glucose (mg dl⁻¹) | 15 | 25 | 27 | 32 | 35 | 37 | 44 | 32 ± 2 | 28 | 31 | 29.5 ± 1.5 |
| Trophamine supplied (g kg⁻¹ day⁻¹) | 6 | 3.2 | 6.1 | 5.8 | 6.3 | 5.2 | 4.9 | 6.0 ± 0.6 | 8.8 | 7.7 | 8.3 ± 0.6 |
| BUN (mg dl⁻¹) | 20 | 16 | 30 | 28 | 33 | 35 | 35 | 28 ± 2 | 23 | 22 | 22.5 ± 0.5 |
| Lipid supplied (g kg⁻¹ day⁻¹) | 0 | 0.19 | 0.24 | 0.23 | 0.17 | 0.15 | 0.20 | 0.20 ± 0.01 | 0.20 | 0.18 | 0.19 ± 0.01 |
| Total calories supplied (kcal kg⁻¹ day⁻¹) | 60 | 49.3 | 73.8 | 69.9 | 72.5 | 57.3 | 57.4 | 72.1 ± 6.6 | 91.3 | 105.0 | 98.2 ± 6.9 |
| Growth (g kg⁻¹ day⁻¹) | NA | 20.2 | 22.0 | 25.7 | 20.0 | 23.8 | 14.1 | 21.7 ± 1.5 | 20.2 | 27.8 | 24.0 ± 3.8 |

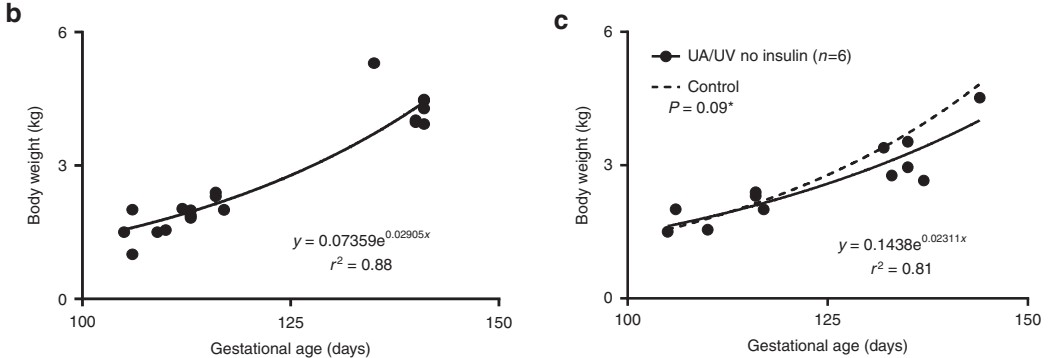

**b** $y = 0.07359e^{0.02905x}$, $r^2 = 0.88$

**c** UA/UV no insulin ($n = 6$); Control; $P = 0.09$*; $y = 0.1438e^{0.02311x}$, $r^2 = 0.81$

**Figure 5 | Nutrition and growth in UA/UV lambs.** Data from $n = 8$ lambs. (**a**) Nutritional substrate and laboratory parameters; data are presented for individual animals and as group averages (mean ± s.e.m.). (**b**) Control growth curve using UA/UV and control lamb weights at hysterotomy ($n = 19$). Solid line represents exponential best fit. (**c**) UA/UV growth curve. Solid line represents exponential best fit of UA/UV non-insulin lamb weights at hysterotomy and at end of studies. Dashed line represents control growth curve. †*In utero* control values in (**a**) derived from measured data in refs 36,37,59. *P value in (**c**) refers to between-group analysis of variance (ANOVA; statistical significance defined as $P < 0.05$).

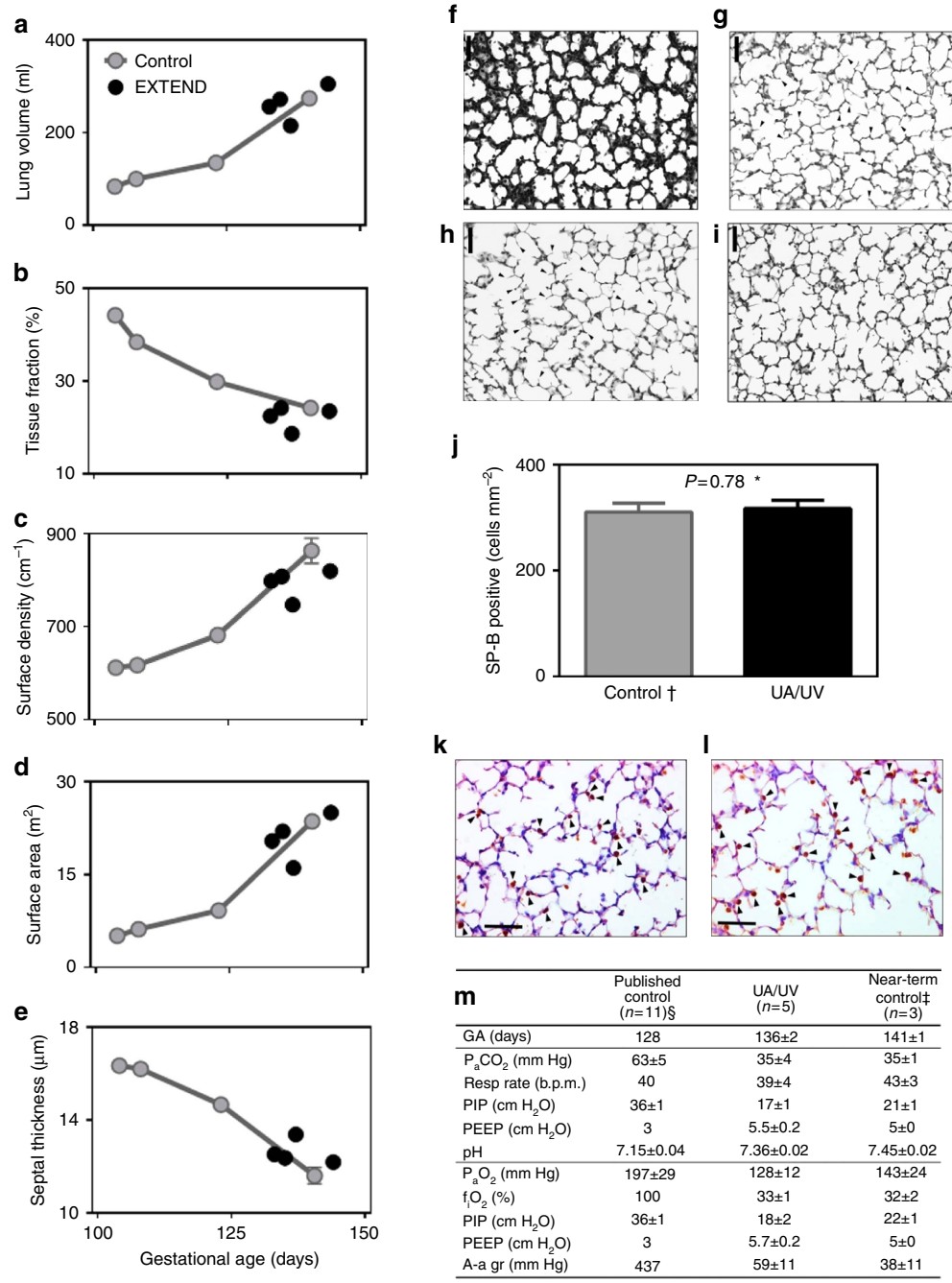

**Figure 6 | Structural and biochemical lung development and early neonatal pulmonary function in UA/UV lambs.** (**a–e**) Morphology of control ((**f**), 113d GA and (**g**),139d GA) and experimental lambs ((**h**), 132d GA, CA/UV lamb 3; (**i**), 144d GA, UA/UV lamb 5) following 19 and 28 days on circuit, respectively (scale bars, 50 μm). Ongoing alveolar formation on circuit is evidenced by increased density and height of secondary septae (arrowheads). Density of surfactant protein-B-positive alveolar cells ((**j**), arrowheads in **k**,**l**) and neonatal pulmonary function were not different from that of age-matched controls (**m**). *P value in (**j**) refers to difference between groups (Student's unpaired t-test, statistical significance defined as P < 0.05). †Control group in (**j**) includes near-term lambs only (n = 4, mean GA 141 days). §Mean control values in (**m**) derived from measured data in ref. 24. ‡Near-term control lambs in (**m**) delivered by caesarean section and ventilated in the same manner as experimental animals. Data in (**j**,**m**) are presented as mean ± s.e.m.

| m | Published control (n = 11)§ | UA/UV (n = 5) | Near-term control‡ (n = 3) |
|---|---|---|---|
| GA (days) | 128 | 136±2 | 141±1 |
| $P_aCO_2$ (mm Hg) | 63±5 | 35±4 | 35±1 |
| Resp rate (b.p.m.) | 40 | 39±4 | 43±3 |
| PIP (cm $H_2O$) | 36±1 | 17±1 | 21±1 |
| PEEP (cm $H_2O$) | 3 | 5.5±0.2 | 5±0 |
| pH | 7.15±0.04 | 7.36±0.02 | 7.45±0.02 |
| $P_aO_2$ (mm Hg) | 197±29 | 128±12 | 143±24 |
| $f_iO_2$ (%) | 100 | 33±1 | 32±2 |
| PIP (cm $H_2O$) | 36±1 | 18±2 | 22±1 |
| PEEP (cm $H_2O$) | 3 | 5.7±0.2 | 5±0 |
| A-a gr (mm Hg) | 437 | 59±11 | 38±11 |

An additional disadvantage that has been raised is parental perception of having their fetus in a 'bag'. It is important to consider that the comparator is the extreme premature infant on a ventilator and in an incubator. We feel that parents will be relatively reassured that their fetus is being maintained in a relatively protective and physiologic environment. The clinical device will be designed with many features that should allow the parent to be connected with the fetus including ultrasound,

a darkfield camera allowing real-time visualization of the fetus within its darkened environment and the ability to play maternal heart and abdominal sounds to the fetus. We therefore feel that the advantages far outweigh the disadvantages of exposure of the fetus to the conventional care imposed on the critically preterm infant in the neonatal intensive care unit environment.

Finally, the umbilical cord offers the only physiologic vascular access for extracorporeal support for the fetus. The use of carotid

**Table 2 | Metabolic, haematologic and fluid parameters in UA/UV lambs cannulated at GA 105–111 days and GA 115–120 days.**

| | GA 105–111 | | | GA 115–120 | | |
|---|---|---|---|---|---|---|
| | Days 0–7 | Days 8–14 | Days 15+ | Days 0–7 | Days 8–14 | Day 15+ |
| Total protein (g dl$^{-1}$) | 3.5 | 3.3 | 3.8 | 4.0 | 3.5 | 3.6 |
| Albumin (g dl$^{-1}$) | 2.0 | 1.9 | 2.1 | 2.1 | 1.9 | 1.7 |
| AST (U l$^{-1}$) | 35 | 28 | 27 | 38 | 26 | 30 |
| ALT (U l$^{-1}$) | 5 | 3 | 3 | 10 | 4 | 23 |
| Alk phosphatase (U l$^{-1}$) | 126 | 93 | 117 | 288 | 106 | 66 |
| Total bilirubin (mg dl$^{-1}$) | 1.1 | 2.2 | 2.6 | 1.0 | 0.7 | 1.8 |
| BUN (mg dl$^{-1}$) | 23 | 23 | 24 | 32 | 39 | 33 |
| Creatinine (mg dl$^{-1}$) | 0.71 | 0.77 | 0.78 | 0.95 | 0.91 | 0.84 |
| Glucose (mg dl$^{-1}$) | 26 | 27 | 33 | 44 | 34 | 40 |
| Sodium (mEq l$^{-1}$) | 139 | 139 | 141 | 142 | 139 | 142 |
| Potassium (mEq l$^{-1}$) | 4.1 | 4.0 | 3.9 | 4.1 | 4.1 | 4.4 |
| Chloride (mEq l$^{-1}$) | 104 | 104 | 105 | 104 | 109 | 103 |
| Calcium (mg dl$^{-1}$) | 10.8 | 12.1 | 12.5 | 9.8 | 10.1 | 10.4 |
| Phosphorus (mg dl$^{-1}$) | 7.3 | 4.0 | 4.1 | 5.7 | 7.3 | 6.6 |
| WBC ($\times 10^3 \mu$l$^{-1}$) | 1.1 | 3.8 | 1.8 | 2.5 | 2.6 | 2.5 |
| Neutrophils (per µl) | 370 | 2,539 | 651 | 870 | 1,600 | 100 |
| Lymphocytes (per µl) | 690 | 1,096 | 1,037 | 1,461 | 889 | 1,615 |
| Platelets (per µl) | 403 | 273 | 214 | 262 | 330 | 143 |
| Haemoglobin (g dl$^{-1}$) | 12.4 | 10.2 | 9.1 | 14.3 | 13.1 | 11.5 |
| Iron (µg dl$^{-1}$) | 189 | 234 | 320 | 103 | 298 | 208 |
| Total fluids infused, ml kg$^{-1}$ h$^{-1}$ (mean ± s.e.m.) | | 9.4 ± 0.8 | | | 10.5 ± 0.4 | |

Alk, alkaline; ALT, alanine transaminase; AST, aspartate transaminase; BUN, blood urea nitrogen; WBC, white blood cell.
N values: for GA 105–111, $n = 5$; GA 115–120, $n = 3$.

arterial inflow or any other peripheral artery does not provide normal placental flows. In addition, all other access sites require an operative procedure on the fetus for both insertion and removal of cannulas, require stabilization of cannulas to avoid catastrophic decannulation events and raise concern for compromise of blood flow to the brain or other structures, depending upon the vessel utilized. While many investigators have utilized the umbilical vessels, most have placed their cannulas into the central vasculature to avoid umbilical vascular spasm[7,8,11–15,23–25,45,46]. This is possible in sheep or goats, but not possible and would likely be hazardous for arterial cannula placement in the human fetus due to the tortuous course of the umbilical arteries and the large cannulas required. In contrast, we sought to take advantage of the natural resistance of the umbilical cord to occlusive events by developing an 'end adapter' design for our umbilical cannulation incorporating very short cannulas and a method for securing the cannulas to the umbilical cord. This avoids irritation of the vasculature or turbulence of flow at the end of the cannulas that might induce vascular spasm, erosion, aneurysm formation or thrombosis. It also allows a length of free umbilical cord between the cannulas and the fetus, preventing concern regarding decannulation or positional occlusion. As a result, we have been able to eliminate sedation during our runs and to maintain much more stable flows.

An alternative approach to AV pumped or pumpless systems has recently been developed that utilizes veno-venous ECMO with circuit inflow via the external jugular vein and circuit outflow via the umbilical vein. They report support of extreme premature lambs in relatively stable physiologic condition for up to 1 week[49–51] with average circuit flows of 87.4 ± 17.9 ml kg$^{-1}$ min$^{-1}$. They also report the requirement for vasopressors for the first 3 days, sedation throughout the run and findings of closure of the ductus venosus on necropsy, with the development of ascites and pleural effusions. In addition, five of nine lambs died before 1 week due to catheter-related complications or arrhythmia. In contrast to our closed fluid environment, an amniotic fluid-filled endotracheal tube is maintained that would not be expected to replicate the normal airway dynamics of fetal breathing. With further development, the veno-venous ECMO system may offer an alternative for support of premature infants, particularly after vaginal delivery as a salvage for ventilatory failure[49].

Clinical application of the technology will require further scientific and safety validation, and evolution and refinement of the device itself. While we report here the application of our system to biologically equivalent premature lambs, the 105-day fetal lamb is considerably larger (1.0–1.5 kg) than an extremely low birth weight premature infant. In pilot studies on size-equivalent but developmentally far less mature lambs (85–95-day GA, 480–750 g), we encountered no limitations related to umbilical vessel caliber or oxygenator flows, confirming the technical feasibility of UA/UV cannulation at this size (Supplementary Movie 5). We did see evidence of too much flow (300–350 ml kg$^{-1}$ min$^{-1}$) in our very early gestational, human-size-equivalent lambs resulting in hydrops and limiting our experiments to 5 to 8 days. In contrast to the more mature and developmentally equivalent lambs, no evidence of autoregulation of circuit flow was observed. This suggests that there is a delicate balance between adequate and excessive circuit flow and that the ability to compensate for increased flow and supraphysiologic right atrial pressures may be dependent on developmental maturity. As the weight-adjusted cardiac output and umbilical arterial size are nearly identical between the fetal lamb and humans, we anticipate the ability to place 8–10 Fr cannulas into human umbilical vessels. We have developed cannulas for rapid insertion and anticipate the ability to cannulate a human umbilical cord and initiate circuit flow in < 2 min that is well within the window of time that is critical for fetal brain oxygenation[52]. This would, at least initially, be done via a modified 'EXIT' procedure[53] in the 50–60% of extreme preterm deliveries that can be anticipated and delivered by caesarean section[54]. The requirement for an EXIT procedure will require centre expertise and introduce additional risk and potential long-term hysterotomy-related morbidity for the mother and these risks will need to be incorporated into

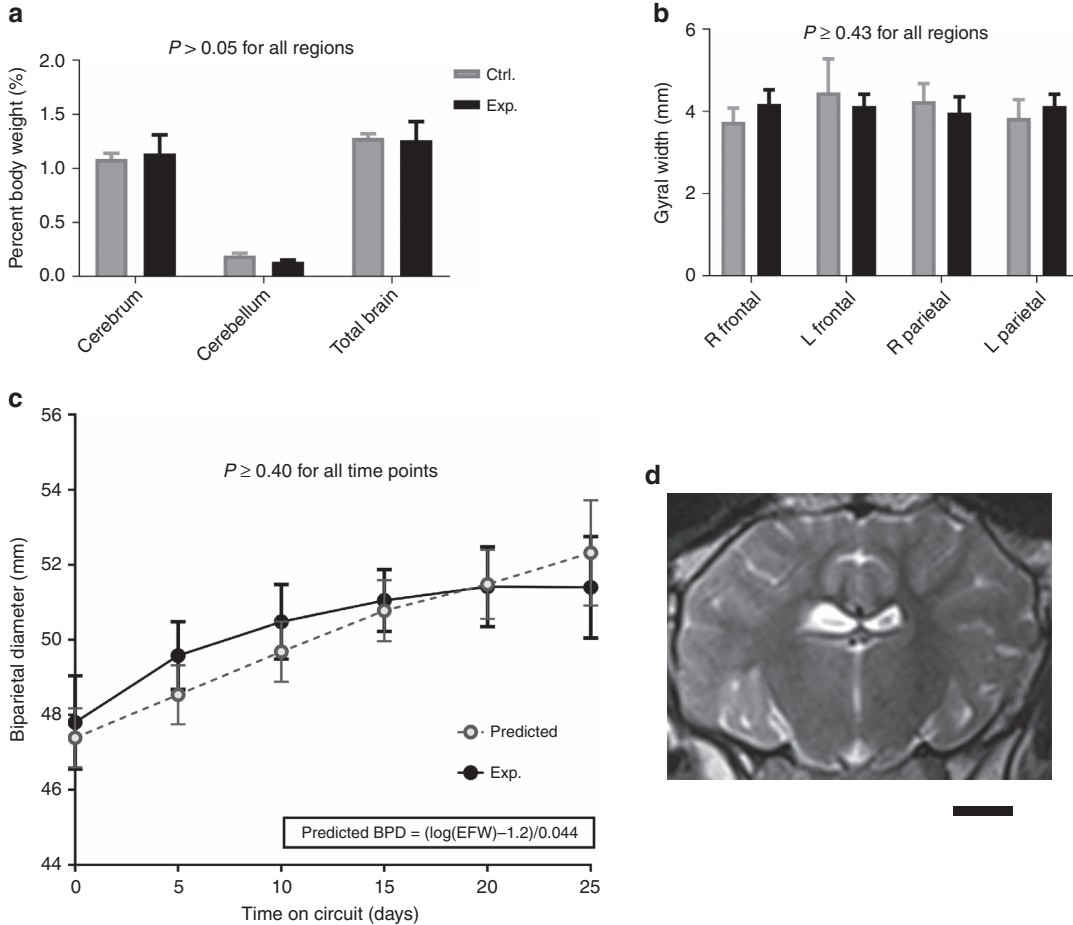

**Figure 7 | Neurologic development and maturation in experimental lambs.** (**a**) Post-mortem brain-to-body weight ratio of experimental (Exp.) and control (Ctrl.) animals; animals that received insulin are excluded. (**b**) Gyral width on haematoxylin and eosin (H&E) stained sections by region as compared with 140-day gestation controls. (**c**) Biparietal diameter in experimental (Exp.) versus predicted (equation derived from ref. 26) animals. (**d**) Postnatal T2-weighted coronal magnetic resonance imaging (MRI) at 6 months of age. Scale bar in (**d**) is 1 cm. Data in (**a–c**) are presented as mean ± s.e.m. P values in (**a–c**) refer to difference between control and experimental groups (Student's t-test for each region/time point, with statistical significance defined as $P < 0.05$). BPD, bronchopulmonary dysplasia.

nondirective counselling before the procedure. The ability to place an infant on the system after vaginal delivery, or to clear contamination under that circumstance, or circumstances of chorioamnionitis, remain to be experimentally determined, but would remain a possibility if umbilical vascular spasm can be prevented or reversed and contamination can be contained and/or fetal infection treated.

The initial clinical target population for this therapy will likely be the 23–25-week extreme premature infant. At the present time the morbidity and mortality of this population would seem to justify general application of this technology if it were proven to dramatically improve outcomes in clinical trials. Future developments may allow better prediction of those infants who are destined for extreme premature delivery[55] and may allow genetic prediction of infants who are most at risk for mortality and morbidity if born premature[56,57]. This would in turn allow risk stratification of potential patients and improved selection of patients who would be most likely to benefit. An important point is that placement on the system would not be prohibitive to standard therapies for extreme premature infants. For instance, prenatal glucocorticoids could still be administered, and rapid delivery from the system and conversion to standard premature neonatal care would be necessary and warranted if the system failed. Before 22–23 weeks of gestation,

there are likely physiologic and technical limitations that will increase the risk and reduce the potential benefit of this therapy. Our goal is not to extend the current limits of viability, but rather to offer the potential for improved outcomes for those infants who are already being routinely resuscitated and cared for in neonatal intensive care units. Finally, the implications of this technology extend beyond clinical application to extreme premature infants. Potential therapeutic applications may include treatment of fetal growth retardation related to placental insufficiency or the salvage of preterm infants threatening to deliver after fetal intervention or fetal surgery. The technology may also provide the opportunity to deliver infants affected by congenital malformations of the heart, lung and diaphragm for early correction or therapy before the institution of gas ventilation. Numerous applications related to fetal pharmacologic, stem cell or gene therapy could be facilitated by removing the possibility for maternal exposure and enabling direct delivery of therapeutic agents to the isolated fetus. Finally, our system offers an intriguing experimental model for addressing fundamental questions regarding the role of the mother and placenta in fetal development. Long-term physiologic maintenance of a fetus amputated from the maternal–placental axis has now been achieved, making it possible to study the relative contribution of this organ to fetal maturation.

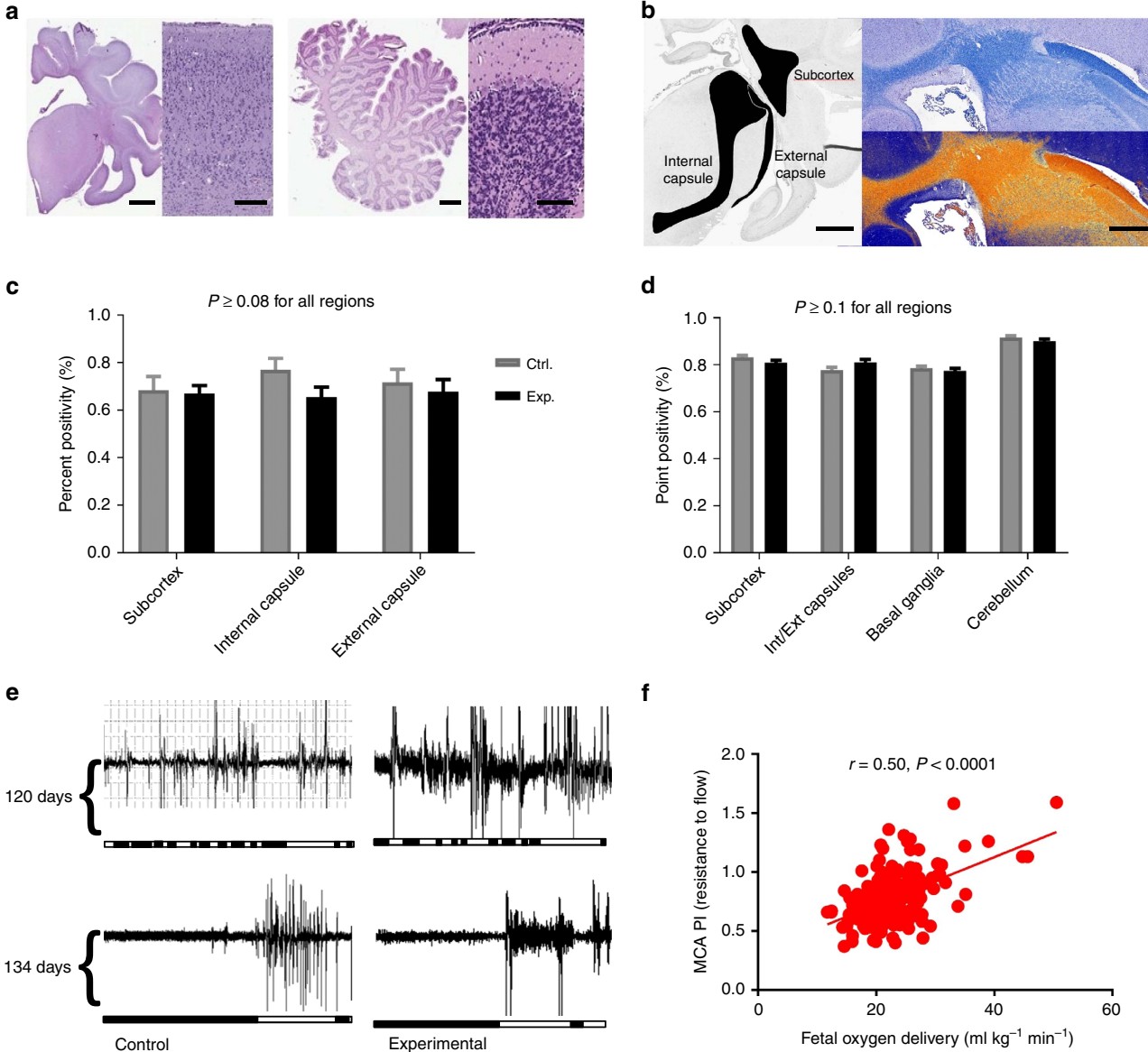

**Figure 8 | Histologic and other parameters of brain development and maturation.** (**a**) Representative haematoxylin and eosin (H&E)-stained sections of post-therapy cerebrum/cerebellum and their respective cortices displaying normal brain parenchyma and absence of injury. (**b**) Representative Luxol fast blue myelin stain and digital pixel identification depicting myelin density (orange). (**c**) Percent positive pixels identified in the selected regions. (**d**) Maximum positivity (density of myelin) in the selected regions. (**e**) Ocular EMG activities in instrumented *in utero* control (Ctrl.) and experimental (Exp.) fetal lambs at 120 days (upper tracing) and 134 and 139 days, respectively (lower tracing). White area of bar represents activity and black areas quiescence. (**f**) Correlation of middle cerebral artery pulsatility index (MCA PI) and fetal oxygen delivery; solid line represents linear best fit. Data in (**c,d**) are presented as mean ± s.e.m. The scale bars in (**a,b**) from left to right are 4 mm, 300 μm, 1.5 cm, 150 μm, 4 mm and 2 mm, respectively. *P* values in (**c,d,f**) refer to difference between control and experimental groups (Student's *t*-test in **c,d** and Pearson's correlation coefficient in **f**, with statistical significance defined as *P* < 0.05).

## Methods

**Surgical procedure.** Time-dated pregnant ewes were used at gestational ages of 104 to 135 days (term is ∼145 days). Animals were treated according to approved protocols by the institutional animal care and use committee of The Children's Hospital of Philadelphia Research Institute.

Ewes were anaesthetized with $15\,mg\,kg^{-1}$ of intramuscular ketamine, with maintenance of general anaesthesia with inhaled isoflurane (2–4% in $O_2$) and propofol ($0.2–1.0\,mg\,kg^{-1}\,min^{-1}$). Intraoperative haemodynamic monitoring included pulse oximetry, with a constant infusion of isotonic saline administered via a central venous line placed in a jugular vein to maintain maternal fluid balance. A lower midline laparotomy was created to expose the uterus, with a small hysterotomy performed to expose the fetal sheep head and neck (CA/JV) or umbilical cord (UA/UV). Experimental lambs undergoing cannulation of the neck vessels (CA/JV and CA/UV) underwent creation of a small right neck incision to expose the jugular vein and/or carotid artery. Fetuses received one intramuscular dose of buprenorphine ($0.005\,mg\,kg^{-1}$). After determination of the maximal

cannula size accommodated by each vessel, ECMO cannulae were placed (8–12 Fr, Medtronic, Minneapolis, MN, USA), with stabilizing sutures placed along the external length of cannulae at the neck. Cannulas were customized with a silicone sleeve over the external portion of the cannulas to permit increased tension of the stabilizing sutures in CA/JV and CA/UV experiments. Experimental lambs undergoing cannulation of the umbilical vessels were positioned to expose the umbilical cord, with connective tissue sharply dissected to expose the umbilical arteries and veins. Umbilical cannulae were placed in one umbilical vein (CA/UV) as well as two umbilical arteries (UA/UV) (12 Fr, Medtronic, or modified 8–12 Fr custom-made cannulas), with stabilizing sutures placed at the insertion sites.

Following construction and blood priming of the oxygenator circuit as described below, connection of the cannulas to the circuit was performed under continuous ultrasonographic visualization of the fetal heart. Occlusion of the umbilical cord was performed immediately following establishment of blood flow through the circuit, with administration of additional blood volume in a subset of animals (CA/JV) demonstrating poor cardiac filling immediately after

establishment of circuit flow. Subsequently, fetal lambs were weighed and transferred to a sterile fluidic incubator for further management as described.

To generate baseline data of the ovine fetus in utero, two time-dated pregnant ewes at 118 days of GA underwent laparotomy for implantation of fetal vascular catheters and electrodes[41]. Insulated multi-stranded stainless steel wire ocular EMG electrodes were implanted subcutaneously in the superior and inferior margins of the muscle overlying the orbit of one eye[41]. Briefly, after induction of general anaesthesia and exteriorization of the fetus, catheters were implanted in the fetal carotid artery and jugular vein, in addition to a reference catheter placed in the amniotic sac, followed by placement of EMG wire electrodes as described above. Fetal catheters and electrodes were exteriorized through the maternal flank, and the uterus and abdomen were closed. Following a 48–72 h recovery period, ewes were transferred to a holding cage for fetal monitoring. Monitoring of catheterized fetal lambs in utero was started 48–72 h after surgery and continued in 24 h intervals on alternating days until completion of the experimental protocol at 140 days of GA. Fetal ocular EMG (1,000 Hz) and arterial pressures (400 Hz) corrected for amniotic fluid pressure were continuously recorded (LabChart 5, ADInstruments Inc., Colorado Springs, CO, USA).

**Circuit.** The pumpless circuit consisted of a low-resistance hollow fibre oxygenator (Quadrox-ID Pediatric Oxygenator, Maquet) connected to ECMO cannulae (Medtronic) or custom-made umbilical cannula via 3/16′ ID × 1/16′ wall thickness BIOLINE-coated tubing (Maquet). In later studies utilizing smaller lambs, a smaller oxygenator was utilized (Quadrox-ID Neonatal Oxygenator, Maquet). Connections were established as an arterial–venous extracorporeal oxygenation circuit, with the carotid artery or umbilical arteries providing inflow to the oxygenator (CA/JV or UA/UV) connected to the oxygenator inflow port and the jugular vein or umbilical vein (CA/JV or UA/UV) providing outflow from the oxygenator and connected to the oxygenator outflow port. Total priming volume was 81 ml of maternal blood for the large oxygenator and 38 ml for the smaller oxygenator. Circuit flow was continuously measured (HT110 Bypass Meter and HXL Tubing Flowsensor, Transonic Systems Inc., Ithaca, NY, USA) and sweep gas supplied to the oxygenator was a blended mixture of medical air, nitrogen and oxygen titrated to achieve fetal blood gas values (target PaO$_2$ 20–30 mm Hg, target PaCO$_2$ 35–45 mm Hg).

**Fluid incubation.** The first CA/JV studies were performed in a 30-litre heated stainless steel reservoir filled with sterile synthetic amniotic fluid ('still reservoir', pilot study), later expanded to a 40-litre polycarbonate tank with continuous recirculation of fluid through a series of sterile filters. Subsequent fluidic incubators were based on a model of continuous exchange of warmed sterile fluid (temperature 38.5–40.5 °C), with inflow tubing mounted on a double-head peristaltic pump and gravitational outflow to facilitate continuous fluid turnover. Fetal lamb enclosures within this system included a 60-litre customized glass tank (CA/JV series one) (NDS Technologies, Vineland, NJ, USA) and individually customized bag enclosures of 2- to 4-litre total volume initially comprised silver-based antimicrobial polyethylene film (CA/UV and UA/UV studies) (Wiman Custom Films & Laminates, Sauk Rapids, MN, USA) and later the same film without silver impregnation. Synthetic amniotic fluid was composed of a balanced salt solution containing Na$^+$ (109 mM), Cl$^-$ (104 mM), HCO$_3^-$ (19 mM), K$^+$ (6.5 mM), Ca$^{2+}$ (1.6 mM), pH 7.0–7.1, osmolarity 235.8 mOsm kg$^{-1}$ water. The rate of fluid inflow (HT110 Bypass Meter and HXL Tubing Flowsensor, Transonic Systems Inc.) and internal fluid temperature (MLT415/A, ADInstruments) were monitored continuously.

**Fetal lamb maintenance on circuit.** Following stabilization and transfer of animals to the fluid incubator, a continuous infusion of heparin (10–400 USP units per hour) and prostaglandin E$_1$ (0.1 µg kg$^{-1}$ min$^{-1}$) were administered intravenously. Heparin dosing was titrated to reach a target activated clotting time of 150–180 s.

Arterial and venous blood were analysed every 1–8 h for blood gas, electrolyte and coagulation values (i-Stat System, Abbott Point of Care Inc., Princeton, NJ, USA) and oxygen saturation (Avoximeter 1000E, Accriva Diagnostics, San Diego, CA, USA). Stored whole maternal blood was transfused as required (10–20 ml kg$^{-1}$) to maintain fetal Hgb levels above 9 g dl$^{-1}$. In a subset of lambs (Prototype IV lambs 4–6), erythropoietin (400 U kg$^{-1}$) was administered intravenously once daily to promote fetal erythropoiesis, and was held for Hgb > 16 g dl$^{-1}$.

Analgesics (buprenorphine, 0.005 mg kg$^{-1}$ intravenously every 3–5 h as needed) and anxiolytics (propofol, 0.1–0.5 mg kg$^{-1}$ min$^{-1}$) were administered during periods of perceived fetal agitation (restless repetitive fetal movements, excessive swallowing, tachycardia and hypertension). This was primarily required in the CA/JV and CA/UV animals with markedly reduced sedation requirement in the UA/UV animals.

Total parenteral nutrition was administered throughout the duration of fetal incubation as described: (CA/JV: amino acids (TrophAmine 10%, 3.5 g kg$^{-1}$ per day), lipids (Intralipid 20%, 2–3 g kg$^{-1}$ per day) and dextrose (10.0–12.5 g kg$^{-1}$ per day) to a total caloric goal of 80 kcal kg$^{-1}$ per day; CA/UV: amino acids (TrophAmine 10%, 3 g kg$^{-1}$ per day), lipids (Intralipid 20%, 1–2 g kg$^{-1}$ per day for lambs 1–3, and 0.1–0.2 g kg$^{-1}$ per day for lambs 4 and 5), dextrose (titrated to blood glucose target 30 mg dl$^{-1}$) and iron (1 mg kg$^{-1}$ per day); UA/UV: amino acids (TrophAmine 10%, titrated to blood urea nitrogen target level 30 mg dl$^{-1}$),

lipids (Intralipid 20%, 0.1–0.2 g kg$^{-1}$ per day), dextrose (titrated to blood glucose target 30–40 mg dl$^{-1}$)and iron (1.0–1.5 mg kg$^{-1}$ per day, titrated to plasma iron target 200–300 µg dl$^{-1}$)).

Cardiac ultrasound was performed one to two times daily by a fetal echocardiographer. Measured parameters included right ventricular (RV)/left ventricular (LV)/combined cardiac outputs (CCO), ductus arteriosus flow and proximal right pulmonary artery pulsatility index. Control echocardiography data were obtained in pregnant anaesthetized ewes at 109 days and 135 days of GA.

**Data acquisition and formulas.** Fetal blood pressure, heart rate, circuit blood flow rates, transmembrane pressure differential, sweep gas flow and incubator fluid temperature were continuously recorded (LabChart 7, ADInstruments Inc.).

Post-membrane oxygen content = (1.34 × Hgb × post-membrane oxygen saturation) + (0.0031 × post-membrane P$_a$O$_2$).

Pre-membrane oxygen content = (1.34 × Hgb × pre-membrane oxygen saturation) + (0.0031 × pre-membrane P$_a$O$_2$).

Weight-adjusted circuit flow = absolute circuit flow/estimated daily weight.

Oxygen delivery (ml kg$^{-1}$ min$^{-1}$) = weight-adjusted circuit flow × post-membrane oxygen content.

Oxygen consumption (ml kg$^{-1}$ min$^{-1}$) = weight-adjusted circuit flow × (post-membrane oxygen content − pre-membrane oxygen content).

Oxygen extraction (%) = (oxygen consumption/oxygen delivery) × 100%.

*Estimated daily weight.* Growth rate was assumed to be exponential and derived from measured body weight at the start and end of each run (according to the formula y = ae$^{bx}$, where 'a' is starting weight and 'b' is growth rate in g kg$^{-1}$ per day). Estimated daily weights (for weight-adjusted calculations) were extrapolated from the exponential growth rate calculated for each lamb.

*Control growth rate.* Initial body weights (at time of delivery from uterus) of Prototype III/IV experimental lambs and late-gestation control lambs were plotted against gestational age. Exponential regression analysis was used to determine control growth rate in utero as well as the estimated fetal weight for the expected biparietal diameter calculations.

**Decannulation and mechanical ventilation.** Following completion of the incubation period, animals were transitioned from the fluid bath, with endotracheal intubation and suctioning to remove excess fluid from the lungs. Surgical decannulation of the carotid artery and/or jugular vein (CA/JV, CA/UV) was performed under general anaesthesia with inhaled isoflurane (2–4% in O$_2$) and propofol (0.2–1.0 mg kg$^{-1}$ min$^{-1}$). Patent umbilical vessels were clamped and divided (CA/UV, UA/UV and control lambs). One umbilical artery or carotid artery was catheterized to enable blood gas measurement. Anaesthesia was then reversed and animals were maintained on mechanical ventilation with intermittent arterial blood gas sampling using an i-Stat System (Abbott Point of Care Inc.). A group of normally grown control lambs (N = 4) were delivered via hysterotomy at 140–141 days of GA, arterial and venous cannula placed for serial blood sampling and fluid/drug administration, respectively, orally intubated and ventilated.

Lambs were maintained on synchronized intermittent mandatory ventilation, with FiO$_2$ titrated to PaO$_2$ > 60–80 mm Hg, peak inspiratory pressure titrated to tidal volume 6–8 ml kg$^{-1}$ body weight, respiratory rate titrated to pH 7.4 (if possible) and positive end-expiratory pressure (PEEP) maintained between 5 and 7 mm Hg. The general goal was to wean ventilator support, and hence lambs were sedated only as needed to optimize respiratory performance, and were often allowed to breathe spontaneously in addition to receiving mandatory ventilator-triggered breaths. Arterial blood gases were obtained every 1–4 h to assess pulmonary gas exchange function.

We could not reliably quantify the degree to which spontaneous, unsupported breaths contributed to oxygenation index (OI = (F$_i$O$_2$ × mean airway pressure)/P$_a$O$_2$) and ventilation efficiency index (VEI = 3,800/(peak inspiratory pressure × respiratory rate × P$_a$CO$_2$)), and hence absolute values of OI and VEI were invalid measures for comparing pulmonary function between lambs in this study. We used relative OI and VEI calculations to determine which blood gas values (at given ventilator settings) corresponded to each animal's 'peak' oxygenation and ventilation (lowest OI was point of peak oxygenation, and highest VEI was point of peak ventilation). Peak oxygenation and ventilation did not necessarily occur at the same time point.

**Post-mortem.** Body, brain and lung weights were recorded. The lungs were inflation fixed (30 cm H$_2$O) via the trachea with 10% formalin. When the fixation pressure had reached a plateau, typically within 20 min, the trachea was occluded and the lungs were submerged in buffered fixative stored at room temperature for 7–10 days. Brains were submerged in 10% formalin for 7–10 days.

**Stereological analysis.** Lung volume (V$_L$) was estimated using the water displacement[58]. Two sections of lung tissue were obtained from the lower lobes, dehydrated through a series of graded alcohol solutions, embedded in paraffin, sectioned at 3 µm and stained with haematoxylin and eosin. Lung images (N = 10 per animal) devoid of major airways and blood vessels were visualized using a Toshiba 3CCD camera interfaced with a Leica DMRD microscope and an

Apple G4 computer. A transparent multipurpose test lattice consisting of 216 test points and a discontinuous series of line probes was placed over the screen images. The number of test points overlying tissue, and number of line probe intercepts with the luminal surface were used to calculate lung tissue fraction and luminal surface density ($S_V$) respectively. Luminal surface area of the lungs ($S_L$) was calculated according to the equation: $S_L = S_V \times V_L$. Septal wall thickness ($T_W$) was calculated according to the following equation: $T_W = V_{pa,tiss}/S_L$; where $V_{pa,tiss}$ is parenchymal tissue volume[25,26].

**Histologic analysis and myelin quantification.** Post-therapy and control brain tissues were sectioned into anterior and posterior regions for each hemisphere. Haematoxylin and eosin and Kluver and Barrera Luxol fast blue staining was performed as detailed in 'Laboratory Methods in Histotechnology' (Armed Forces Institute of Pathology, Washington, DC)[38,39]. Slides were digitally scanned at $20 \times$ magnification and evaluated using both Aperio Imagescope Version 12.3 (Leica Biosystems Pathology Imaging, Buffalo Grove, IL, USA) and standard light microscopy. Individual gyri were digitally measured at maximum width on each slide. Myelin analysis and the calculation of myelin-positive pixel ratios was performed using Positive Pixel Count Version 9 (Aperio Technologies, Leica Biosystems) A uniform algorithm adjusted to identify pixels consistent with positive staining was applied to experimental and control slides. Maximum positivity by region was determined by limiting density analysis to multiple areas $1 \, mm^2$ in size where the Luxol fast blue staining was strongest[40].

**Statistical analysis.** Haemodynamic parameters (heart rate, mean pre-membrane pressure and circuit flow) and arterial blood gas parameters were averaged over 12 and 24 h, respectively, and analysed using one within-group (time) and one between-group (treatment) analysis of variance. When significant differences between means were detected, multiple comparison analysis was performed using least significant difference test (SPSS Version 23, IBM Corp., Armonk, NY, USA). Epithelial SP-B cell density, biparietal diameter, brain-to-body weight ratios, myelination density and fetal breathing responsiveness to $PaCO_2$ were compared using Student's unpaired $t$-test (SPSS). Body weight data were fitted to nonlinear regression curves, the rate constants of which were analysed between groups (GraphPad Prism Version 6, GraphPad Software, San Diego, CA, USA). Significance was accepted at $P < 0.05$. All data are presented as mean ± s.e.m.

**Data availability.** The authors declare that all data supporting the findings of this study are available within the article and its Supplementary Information Files or from the corresponding author on reasonable request. The *in utero* control data in Figs 3–6 were obtained from published work[18,20,24,36,37,59].

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

## Acknowledgements

This work was supported by generous funding from the Children's Hospital of Philadelphia Institutional Development Fund and the Department of Surgery. We gratefully acknowledge the following individuals and organizations for technical support and contributions to this work: Dr Claudius Diez, Clinical Director Cardiopulmonary Surgical Therapies, Maquet Getinge Group; Nicole Gavula; Dr Charles Vite; Dr Elizabeth L. Buza; Dr Brian Harding; Dr Andrew Misfeldt; Dr Zhiyun Tien; Antoneta Radu; Dr Judith Grinspan; Abby Larson; Jenny Kim; Orlando Castillo; Grace Hwang; Kathleen Young; NDS Technologies, Inc.; Russen Brothers Construction Inc.; and The Comparative Pathology Core of the University of Pennsylvania.

## Author contributions

E.A.P, M.G.D. and M.A.H. performed all experiments and participated in their design and interpretation. E.A.P., M.G.D., and A.W.F. shared in device conception, system design and implementation. M.G.D. was the primary engineer and builder of the device. E.A.P., M.G.D., M.A.H., A.Y.M., P.E.M., J.D.V., C.M-B., A.O., A.J.S. and R.C.C. provided experimental support, managed lambs on the device and performed data acquisition. M.G.D. performed and interpreted pulmonary morphometry and histology. J.H. and J.R. performed echocardiography, provided cardiology expertise and interpreted echocardiographic data, T.R.W. and J.T.C. provided essential technical and experimental support. A.W.F., W.H.P. and H.L.H. developed and performed all surgical procedures. K.C.D. provided neonatology and physiologic expertise, and participated in neonatal resuscitation and respiratory data acquisition. A.W.F. led the project, provided scientific direction and wrote the paper with support from E.A.P., M.G.D. and M.A.H.

## Additional information

**Competing interests:** The authors declare no competing financial interests.

