## [Peer Review File · Nature Communications]

Reviewer's Comments

Reviewer #1 (Remarks to the Author)

The major claim of this paper is the ability to support extremely premature lambs "in an extra-uterine device maintaining normal or near-normal development for up to four weeks." The authors present substantial data detailing their evolution of a pumpless extracorporeal system. Overall, the paper is well written and would be of interest to others in the field.

The authors refer to this technology as an "extra-uterine device" or an "extracorporeal device." While both terms are appropriate, I will use the term "artificial placenta" in my review as this is the most accepted term in the literature. The concept and research history of the artificial placenta dates back nearly 60 years. The idea of supporting premature infants with extracorporeal support without mechanical ventilation was conceived shortly after the advent of cardio-pulmonary bypass for heart surgery. Many investigators have tried to recreate the intrauterine environment with an extracorporeal artificial placenta, but as Partridge et. al. points out, they have met with episodic and limited success. Many research efforts were not sustained, and historically the technology and knowledge of both fetal physiology and the complications of extreme prematurity were rudimentary. Nevertheless, a number of investigators have published long term survival of premature sheep or goats using an extracorporeal artificial placenta for 1 to 3 weeks with varying degrees of hemodynamic, cardiac and gas exchange data.

While the overarching claims of this paper (long-term extracorporeal support of extremely premature lambs) are not novel per se, in terms of duration of extracorporeal support, the authors' results in six extremely premature lambs are outstanding, demonstrating 4 weeks of support with hemodynamic stability, maintenance of fetal circulation, and excellent gas exchange. The supporting hemodynamic, cardiac, and gas exchange data are thorough and well presented. Other noteworthy advances in this field are demonstrating lung maturation and function, achieving sterility with their 'Biobag' device, and eliminating umbilical arterial and venous spasm.

The authors present varying levels of data on organ maturation and protection. The lung data presented is comprehensive and includes detailed histologic analysis, surfactant levels, and physiologic data with mechanical ventilation after being weaned from extracorporeal support. This lung data is further strengthened by appropriate age matched controls. The lung data overall support the authors' conclusion. However, the data on brain protection are quite preliminary and strong conclusions cannot be made. The authors present qualitative neurologic assessment, one MRI in a survivor, a few EEG tracings and a few representative histologic brain sections. It should be noted that the lamb is not a good model of intraventricular hemorrhage since the germinal matrix is present at approximately 70 days gestation. Therefore, the lack of intraventricular hemorrhage despite anticoagulation with heparin in this system does not indicate that this problem is solved. Nevertheless, the sheep model has been used to assess white matter injury and further MRI data with targeted histologic analysis of pathologic areas (with gestational controls) would provide further evidence of brain protection. Given the stable perfusion data presented, it may be inferred that cerebral perfusion is adequate but this needs to be proven with NIRS or carotid flow probe data in subsequent work. Therefore, the conclusions about brain protection need to be modest.

The authors use commercially available components to construct their extracorporeal circuit including the Maquet Quadrox (pediatric and neonatal) oxygenator, 8-12F Medtronic cannulas, and BIOLINE® -coated tubing. While the concept of an 'amniotic bath' has been described and used many times in this context, the authors data suggest that they have made substantial progress addressing infection with their silver impregnated and filtered 'Biobag.' Although this is an appealing strategy for incubation and lung management as it most closely replicates the actual intrauterine environment, the potential disadvantages should be pointed out in the discussion. For

example, infection is still an issue if caregivers need to access the 'Biobag' and perhaps more importantly, it would make access to the fetus difficult especially in emergent situations.

The discussion is unusually short and the authors' claims are not appropriately discussed in the context of previous literature. Of the four noteworthy advances (long duration of support, evidence of lung maturation and function, 'Biobag', and elimination of umbilical vessel spasm) there is a body of work in the literature which needs to be discussed and referenced. If space is a limitation, I would suggest removing some of the data on the evolution of these experiments and focus on the umbilical artery / vein access animals and provide a robust discussion.

As the authors point out in their discussion, the utility of their extracorporeal device (AP) is potentially manifold. First, it supports the feasibility of long-term extracorporeal support in an extremely premature lamb model with preliminary results suggesting organ maturation and protection. Second, in its current iteration, it may be useful to study fetal physiology.

As the authors point out, an extracorporeal artificial placenta has its greatest potential utility as a means of supporting extremely premature human infants. In this regard, the authors need to elaborate on the feasibility of this design and approach to clinical translation. The advantages and disadvantages of other modes of extracorporeal support such as the veno-venous approach using the umbilical vein for infusion and the jugular vein for drainage should be discussed. As the authors point out, they 'maintain a length of native umbilical cord (~5cm), use papaverine, atraumatic operative technique, and maintain warmth and physiologic oxygen saturation of the umbilical venous inflow on initiation of circuit flow.' It appears these steps are critical to prevent umbilical vessel spasm and allow the pumpless system to work properly. This procedure is meticulous and while it can be performed in an experimental situation, the authors need to discuss how this would occur in the clinical scenario. Premature birth is usually precipitous, the cord is divided and the infant is intubated with high oxygen delivery. In this scenario, the umbilical vessels are in spasm and although umbilical lines are routinely placed, they are very small and could not support a pumpless extracorporeal system. The authors should clarify if an ex utero intrapartum treatment (EXIT) procedure would be required for clinical translation of this device. If so, it should be noted that this would require special expertise at a fetal therapy center, knowledge of impending premature delivery, additional maternal risk, and no risk stratification for the infant apart from extreme prematurity. Although the overall outcomes of extremely premature infants is poor, there are some infants who survive with good outcomes using less invasive ventilator strategies and do not need invasive treatment. It should be noted that although the authors successfully supported even smaller lambs, they have large umbilical vessels which would accommodate 8-12F cannulas. In this paper, based on the large cannulas, short length of tubing, low resistance oxygenator, and absence of umbilical vessel spasm, the flow through the device was approximately 25%-35% of the lambs combined cardiac output which allowed for excellent long-term support without heart failure. Since the normal combined cardiac output of a human fetus is similar to the sheep (~500-600 ml/kg/min), the authors should address feasibility of extracorporeal support using much smaller cannulas which would dramatically increase resistance and limit flow even in the absence of vessel spasm.

Lastly, as it relates to clinical translation, the authors should mention the unsolved problem of anticoagulation in extracorporeal support. Extremely premature infants are at high risk for intraventricular hemorrhage and strategies need to be developed to avoid anticoagulation in an artificial placenta.

Reviewer #2 (Remarks to the Author)

The report represents a description of development towards a system that can harbor a premature fetus and allow this to grow ex utero. Premature birth and extreme premature birth is a large health problem so the clinical significance is clear.

There are numerous previous reports in sheep and goat addressing the same problem, but with

the presented paper showing a more sophisticated system with possibilities for longer duration of supported development ex utero.

The data is well presented and the authors have looked at a large number of parameters that indicate that the ex utero development is similar to the in utero development during this time of gestation.

The major problem with the study is that the system does not seem to be optimized yet and that the authors do not present a larger series of animals in fully optimized machine.

I understand that only one animal survived so that it could be without ventilation. Moreover, bacterial contamination is seen in AF in all the experiments and in blood in many. .

The authors should address these points

- larger number of animals included
- set, and clinically relevant, age for removal of the lamb from the machine
- a consistent high success rate should be reached
- appropriate control group (in utero developed fetuses) of same age
- long follow up of lambs delivered from the system
- consistency

Reviewer #3 (Remarks to the Author)

The goal of the work presented in this publication was to develop a system to support a "premature" ovine fetus. The investigators tested the hypothesis that a pump-less circuit with blood flow driven by the fetal heart combined with a low resistance oxygenator would mimic the normal fetal/placental circulation. The volume of the system was devised to be comparable to the blood volume of the placenta.

The results section describes initial attempts to achieve oxygenation by various means, which included arterial inflow from the common carotid artery and venous return via the right jugular vein. They encountered many problems including sepsis, thrombus formation and exsanguination. Due to the problem of bacterial overgrowth, they developed a biobag design, a closed system that solved the contamination problem and eliminated fetal pneumonia.

The next iteration of the system was carotid artery/umbilical vein cannulation. They found that flow to the oxygenator was restricted due to the small caliber of the artery.

Therefore, they moved to double umbilical artery and single umbilical vein cannulation. With this approach they were able to maintain fetuses up to 28 days at which time the experiments were terminated. Essentially all the physiological parameters they assayed were very close to those measured in utero, suggesting that this method has real promise. Growth was comparable to breed-matched controls. Lung development appeared normal as assessed by histology, surfactant B expression and functional analyses after birth. Bilirubin levels and liver function tests were either within the normal range or mildly elevated. Neurologically the animals appear to be intact according to several criteria including ocular electromyography. An MRI, which was performed on one surviving animal, was normal. Upon long-term follow-up, this animal had normal neurologic behavior.

The authors present very interesting data in which they used a sheep model to develop a new type of incubator for premature fetuses that more closely resembles the intrauterine environment. The results are interesting and the approach innovative. A significant portion of the results section describes experiments that did not work. This part of the manuscript could be reduced to a brief summary of what was tried before attempts were successful.

Reviewer #1

We thank this reviewer for the historical perspective provided and the generally favorable comments and helpful suggestions.

- 1) *However, the data on brain protection are quite preliminary and strong conclusions cannot be made. The authors present qualitative neurologic assessment, one MRI in a survivor, a few EEG tracings and a few representative histologic brain sections.*

The reviewer is correct regarding the limited neurologic assessment in our study. The model is limited by poor survival of premature lambs when transitioning off of extracorporeal support. This is a common problem in studies utilizing the lamb model in that there are a number of lamb specific issues that limit long term survival. For instance bowel distention and ileus that compromise ventilation, gastrointestinal hemorrhage, immune issues (lack of colostrum), and complications of mechanical ventilation. This is reflected in failure of survival of even our age matched controls after cesarean delivery, sedation, and mechanical ventilation. We simply do not have equivalent capability to the human NICU for premature lamb resuscitation and maintenance. Thus we only have one survivor to assess long term neurologic outcomes. In reality, the ability to meaningfully assess neurologic outcome in sheep is also limited. Even the FDA has agreed after discussion related to IDE application, that long term neurologic follow up is not required in our model

In addition to our previous limited neurologic assessment (which is qualitatively normal) we have extended our description and analysis of the brain in the 6 UA/UV animals reported in our original manuscript and an additional 2 animals run for 672 and 627 hours. These assessments include more detailed assessment of brain growth (brain (cerebrum and cerebellum) to body weight ratios, and bi-parietal diameter profiles during support as a surrogate of brain growth compared to age matched controls. We also formally assessed gyral width as a measure of brain maturation relative to age matched controls. These parameters of growth and development are shown in the new Figure 7 along with the MRI from our single survivor. We were unable to perform perfusion fixed MRIs on the brains of our fetuses due to practical limitations but hope to do this in a future study.

- 2) *Nevertheless, the sheep model has been used to assess white matter injury and further MRI data with targeted histologic analysis of pathologic areas (with gestational controls) would provide further evidence of brain protection.*

We agree that white matter is most susceptible to ischemic injury and we performed targeted histologic analysis of white matter by performing Luxol fast blue myelin staining of critical brain regions (Internal capsule, External capsule and Subcortex). These were assessed along with H&E stains by two pathologists for any evidence of myelin abnormalities relative to age matched controls. We also performed a pixel based densitometry analysis of myelin staining in each region versus age matched controls. These results are shown in the new Figure 8. The conclusion of this more detailed histologic analysis for white and gray matter was that no abnormalities were noted in either gray or white matter development.

- 3) *Given the stable perfusion data presented, it may be inferred that cerebral perfusion is adequate but this needs to be proven with NIRS or carotid flow probe data in subsequent work. Therefore, the conclusions about brain protection need to be modest.*

We have added data (Figure 8) on middle cerebral artery pulsatility index demonstrating evidence of normal autoregulation of cerebral blood flow in response to oxygen delivery. We agree that this, in combination with our data on normal

cardiac output, fetal circulation, and oxygen delivery strongly supports normal cerebral perfusion but only indirectly. Unfortunately, we have been unsuccessful thus far in adapting NIRS technology to our lambs. We have attempted to apply a brain tissue pO₂ probe in three lambs (not included in this study). Unfortunately while spot measurements of tissue pO₂ have been in the normal range, we have not been able to keep the probes functioning long enough to get sustained measurements. Thus we still have only indirect measurements of cerebral perfusion/oxygenation. We have tempered down our conclusions on brain protection accordingly (see the statement below).

- 4) *It should be noted that the lamb is not a good model of intraventricular hemorrhage since the germinal matrix is present at approximately 70 days gestation. Therefore, the lack of intraventricular hemorrhage despite anticoagulation with heparin in this system does not indicate that this problem is solved.*

We agree with the reviewer and have directly stated this limitation in the revised manuscript at the end of the results section. "Taken together with our observation of normal cardiac outputs, oxygen delivery, and fetal circulatory pathways, we feel our limited data to this point is encouraging with respect to cerebral perfusion and brain development. It is important to note however, that there are important differences between fetal lamb brain maturation and human brain maturation, most important of which is the earlier maturation of the germinal matrix in the lamb (70 days)²⁸. Thus, the risk of intracranial hemorrhage cannot be assessed in the ovine model²⁹. In addition, long term neurologic follow up is difficult in our model, due to difficulties with survival of premature lambs, and to the limitations in assessment of lamb neurologic function. Thus any conclusions regarding neurologic development must be qualified."

- 5) *Although this is an appealing strategy for incubation and lung management as it most closely replicates the actual intrauterine environment, the potential disadvantages should be pointed out in the discussion. For example, infection is still an issue if caregivers need to access the 'Biobag' and perhaps more importantly, it would make access to the fetus difficult especially in emergent situations.*

We have addressed this request in the discussion. "Disadvantages of a fluid environment include limited access to the fetus by caregivers (i.e. physical examination, blood draws, hemodynamic monitoring) with contamination of the environment if such access is required, and lack of rapid access in the case of an emergency. In our system, we can examine the fetus in great detail by ultrasound which is superior to physical examination, and can access the circuit for all our vascular access needs (hemodynamic monitoring, blood draws and fluid/nutritional support). Sterile access ports for suction of meconium and other purposes are included, and we have designed a clinical device that could be rapidly opened should the fetus need to be resuscitated. We therefore feel that the advantages far outweigh the disadvantages of exposure of the fetus to the conventional care imposed on the critically preterm infant in the NICU environment."

- 6) *The discussion is unusually short and the authors' claims are not appropriately discussed in the context of previous literature. Of the four noteworthy advances (long duration of support, evidence of lung maturation and function, 'Biobag', and elimination of umbilical vessel spasm) there is a body of work in the literature which needs to be discussed and referenced. If space is a limitation, I would suggest removing some of the data on the evolution of these experiments and focus on the umbilical artery/vein access animals and provide a robust discussion.*

We thank all three reviewers for this constructive criticism. We have abbreviated the previous discussion of prototype development into a single section (Pilot Studies). We have focused the results section on the 8 animals supported by the final device and extensively expanded the discussion as suggested placing our work in the context of previous studies

- 7) *As the authors point out, an extracorporeal artificial placenta has its greatest potential utility as a means of supporting extremely premature human infants. In this regard, the authors need to elaborate on the feasibility of this design and approach to clinical translation.....*

We have included significant discussion related to this point and subsequent points raised by this reviewer, all of which are legitimate.

This procedure is meticulous and while it can be performed in an experimental situation, the authors need to discuss how this would occur in the clinical scenario. The authors should clarify if an EXIT procedure would be required for clinical translation of this device,. If so, it should be noted that this would require special expertise at a fetal therapy center, knowledge of impending premature delivery, additional maternal risk, and no risk stratification for the infant apart from extreme prematurity.

As requested, we have discussed our strategy for delivery by a modified EXIT procedure. While an EXIT approach will be employed, we have developed and tested cannulas for human use that can be inserted very rapidly with institution of circuit flow within 2 minutes of cord exposure. We think that this will minimize the anesthetic requirements relative to a conventional EXIT procedure. The need for center expertise for an EXIT, and its additional attendant risk are now mentioned in the discussion.

While at the present time, risk stratification for extreme premature infants is not possible, it will likely be possible in the future as this reviewer points out. This will only improve our ability to select appropriate infants for delivery and support. However at the present time, the morbidity and mortality of this population as a whole is severe enough to justify no risk stratification, if it proves to dramatically reduce morbidity and mortality. This point has been added to the discussion.

The authors should address the feasibility of extracorporeal support using much smaller cannulas which would dramatically increase resistance and limit flow even in the absence of vessel spasm.

To address the relative size difference between the premature lamb at developmental equivalence to the human we describe pilot studies that we have done (with a supportive video) demonstrating feasibility of cannulation of a less than 500 gm lamb (equivalent to 17 weeks gestation developmentally). The umbilical vessels are equivalent or smaller (2 UVs instead of one) in size to human umbilical vessels at the targeted gestational age (23-25 weeks). There were no limitations with cannulation (8 – 10 Fr cannulas), or with running the animals for up to 8 days on our circuit. The problem was not too little flow but too much flow as described in the discussion. Thus, with due respect to this reviewer, if the umbilical vessels are not in spasm, they are more than large enough to accommodate the cannula sizes required for physiologic or even supra physiologic flow. We do not feel that our system as currently envisioned could be applied after vaginal delivery and after cord spasm has developed. If cord spasm could be prevented or reversed, than it may be a possibility, assuming contamination can be cleared or controlled in the system. These points are addressed in the discussion.

With respect to anticoagulation, we have added discussion on the potential risk of anticoagulation in our system and its potential impact on the high risk of intracranial hemorrhage. This is a difficult question to answer due to the evidence that ICH is highly related to positive pressure ventilation, use of pressors, and other variables in the current NICU environment, all of which would be avoided by use of our system. We also as requested have discussed the veno-venous approach with its advantages and disadvantages. We hope that we have satisfactorily addressed the reviewers points within the limitations of space and scope of this manuscript.

Reviewer #2

1) The major problem with the study is that the system does not seem to be optimized yet and that the authors do not present a larger series of animals in fully optimized machine.

We apologize for the structure of our previous manuscript describing in detail the evolution of the final device. This may have given the impression that our final device was still in evolution. In fact it is fully optimized for the sheep model and we are developing a human prototype based on the sheep device. I would ask this reviewer to examine the current manuscript and note that the device has not been changed from its final form for the 6 UA/UV animals presented in the first manuscript or the additional two animals supported for 672 and 627 hours in this revision. This constitutes 8 animals supported on the same device for 468 – 672 hours. The 6 animals in the original manuscript were consecutive animals. The additional 2 were run in response to this reviewer's request for more animals and as we have been performing other studies to look at metabolic and other parameters these were not consecutive with the first 6. As these runs are extremely resource and labor intensive I feel that 8 animals are a large number relative to other studies of this nature and hope that the reviewer agrees.

2) I understand that only one animal survived so that it could be without ventilation. Moreover, bacterial contamination is seen in AF in all the experiments and in blood in many.

As addressed in my initial statement above the reviewer is correct that only one animal has survived long-term from this study. However, all of the animals in the UA/UV group survived for transition to mechanical ventilation and all had good gas exchange with the exception of the animal with TOF. There is abundant evidence in the literature supporting difficulty in achieving survival of premature ventilated lambs and no studies of long term survival of premature lambs after ventilated support that I am aware of. This was confirmed by our inability to achieve independence from the ventilator in even our age matched controls delivered by cesarean section and ventilated. With respect to bacterial contamination this was limited to low level contamination of amniotic fluid cultures that appeared late in the runs in all of the UA/UV biobag sheep. There were no positive blood cultures in this group. The contamination is not surprising given the relative lack of sterility in a sheep facility and the necessity in some of the sheep to enter the bag (for instance to place a vesicostomy in male sheep with urinary obstruction due to urachal occlusion, or for minor bleeding at the site of urachal catheter insertion). As stated in the manuscript there have been no clinically significant infections in the bag and the low level contamination could be cleared or controlled by the addition of antibiotics and increasing the exchange rate of fluid.

3) Larger number of animals included.

As above, our revision includes two additional animals in addition to the original 6

supported on the same device. We feel this is an adequate number given the limitations and expense of the model.

4) *Set, and clinical relevant, age for removal of lamb from the machine.*

With respect to the reviewer we are not sure what is meant by clinically relevant. As per our study design, we place lambs on at an age developmentally analogous to human infants at 23-26 weeks gestation (105-113 days Gestation) and run 21 - 28 days from there. The lamb comes off either when evidence of reduced oxygen delivery requires us to remove the animal from the machine, or when our animal protocol limitation (28 days) is reached. If the reviewer wishes us to start later in gestation to accomplish survival, that would actually be less clinically relevant to the extreme premature infant.

5) *a consistent high success rate should be reached*

Again, with respect to the reviewer, we have demonstrated that we can consistently achieve 21 to 28 day survival in extreme premature lamb fetuses. This is evident from our 6 consecutive lambs reported in the original manuscript and the 2 additional lambs run for the purpose of adding numbers since our original manuscript. We have not attempted other lambs that have failed in this series. That is successful in the context of our original experimental goals and in our view consistent.

6) *appropriate control group (in utero developed fetuses) of same age*

We have included age matched in utero control lambs at every opportunity. This includes analysis of pulmonary morphometrics and respiratory function, brain development and histology, growth, EMG experiments, and echocardiographic measurements. Where appropriate we have also included control data from the literature for comparison.

7) *Long follow up of lambs delivered from the system.*

Although we would love to have surviving lambs for long term follow up this, for the reasons stated above, is not a realistic goal in our model system. We would need to invest a huge effort in a premature lamb intensive care unit for probably low yield given the general inability to achieve this goal by ourselves or other investigators. While review of the literature reveals a vast majority of acute studies in premature lambs, there are a few studies of long term ventilation (up to 3 weeks) of 132 day gestation lambs to induce BPD, but these are associated with high loss rates and no survival studies have been reported. Review of the long term ventilation studies is illustrative of the difficulties of managing preterm lambs¹. In a study in which lambs had induced vaginal deliveries at 133 days and did not receive sedation and were cared for by the mother, survival was only 60% and 44% for males despite addition of mask CPAP for hypoxemia². The added effects of sedation and cesarean delivery (or in our case delivery from the biobag) impact survival dramatically as confirmed by our inability to achieve survival after cesarean delivery and mechanical ventilation (requiring sedation) of normal control lambs. Even in discussion with the FDA, they have waived this requirement due the difficulty, impracticality, and uncertain endpoint analyses for neurologic development. We respectfully ask this reviewer to reconsider this request.

1. Null DM, Alvord J, Leavitt W. et al. High-frequency nasal ventilation for 21 d maintains gas exchange with lower respiratory pressures and promotes alveolarization in preterm lambs. *Pediatr Res* 2014, 75:507-16.
2. De Matteo R, Blasch N, Stokes V, Davis P, Harding R. Induced preterm birth in

sheep: A suitable model for studying the developmental effects of moderately preterm birth. *Reproductive Sciences* 2010, 17:724-733.

8) *Consistency*

See above

Reviewer #3

We appreciate this reviewer's favorable comments. We have significantly revised the manuscript as outlined above and reduced the early experiments to a short "Pilot Studies" section with expansion of the discussion.

Reviewers' Comments:

Reviewer #4 (Remarks to the Author)

The revised manuscript has largely answered the reviewers' questions and comments. The addition of 2 extra animals adds significantly more data. Also the additional post mortem brain studies suggest there was no significant neurological damage. However, this analysis was possible in just a single individual so it is not possible to draw any firm conclusions. Lung function appears relatively normal. These data show great promise for successful extracorporeal maintenance of infants.

However, I have some additional concerns. There are very few 23 weeks' human infants who survive and of course survival increases as gestation increases. Many 24 weeks' infants do survive with current NICU management. Have the authors discussed their system with any parents of preterm infants? Although the data look promising, the videos of the lambs inside the biobags would be quite confronting to people not familiar with such things. I wonder how parents would react to the prospect of the neonatologist placing their baby in what looks like a plastic bag full of fluid. They are already highly stressed because their baby has been delivered at such a premature stage.

Glucocorticoids are usually administered to women with threatened preterm labour if delivery can be delayed >24h later. Their purpose of course is to mature the baby's lungs and other organs in preparation for extrauterine life. Given that most extremely preterm babies will have been exposed to maternal glucocorticoids, how will this affect an infant's ability to be EXITED and placed inside a biobag? Given over 40 years of glucocorticoid therapy to prevent RDS in preterm infants the concept of not using glucocorticoid therapy would be difficult to "sell" to clinicians.

Many extremely preterm infants have been exposed to intrauterine infection which needs treatment. However, there are often long term consequences of that infection and the neurological damage that ensues is of particular concern. Are there interventions that would be provided to such infants that would not be possible if they were inside a biobag?

Amniotic fluid is composed of a lot more than the relatively simple synthetic amniotic fluid that has been used here. It is a cocktail of hormones and growth factors as well as salts and nutrients derived from maternal tissues, the amnion and fetal urine. These contribute to fetal lung and gut development. Have the authors considered including some of these other factors?

Reviewer #5 (Remarks to the Author)

I was not one of the original three reviewers (although I was invited to be but was unavailable I think) - however I have reviewed the original reviewers comments and the author's responses and think the paper can now be published. I accept that for ethical reasons the authors cannot include longer term or more animals and while this is unfortunate I think the authors have made a pretty good stab at dealing with the major points raised notwithstanding this limitation.

The 'quest' to develop a truly functioning uterus outside a human body has almost become a 'holy grail' in some areas of reproductive biology and it is not hard to see the sorts of imperatives that make it so - not the least that we can't yet even manage to culture uterine epithelium and get it to respond to hormones in the same way as it does in situ.

This paper takes us some way towards a functional extra-corporeal uterus at least for the later stages of pregnancy.

Reviewer #6 (Remarks to the Author)

To the Authors,

The authors have responded very favorably to almost all critiques of their original submission. There are two minor points which were not raised with the original submission that the authors should consider.

One is that the procedure for placement of the UV catheter includes the final location of the tip of the catheters. Similar information for the UA catheters is not given in the text, and a brief mention of this in the text would be helpful.

Second, the authors correctly offer persistence of a fetal circulatory system as a major advance of their method over others. Although the authors describe the use of PGE and documentation of maintenance of the ductus arteriosus, they do not comment on assessment of ductus venosus patency. Perhaps this was assessed during the cardiac ultrasounds? If this was not documented, then they cannot make the statement that use of the umbilical vein inflow mimicked "normal fetal umbilical venous return and improve[d] streaming of oxygenated blood across the foramen ovale." As I am sure the authors are aware, if the DV is closed there will not be any streaming. Beyond this, without the ductus venosus all of the input into the umbilical vein from the circuit as well as all the blood flow through the portal veins will perfuse the liver before entering the IVC, where presumably some hepatic metabolism will take place, as opposed to fetal circulation where much of the UV and portal blood is shunted through the liver.

Reviewer #4

We thank this reviewer for willingness to review the revised manuscript on short notice and for the favorable comments.

- 1) *However, I have some additional concerns. There are very few 23 weeks' human infants who survive and of course survival increases as gestation increases. Many 24 weeks' infants do survive with current NICU management. Have the authors discussed their system with any parents of preterm infants? Although the data look promising, the videos of the lambs inside the biobags would be quite confronting to people not familiar with such things. I wonder how parents would react to the prospect of the neonatologist placing their baby in what looks like a plastic bag full of fluid. They are already highly stressed because their baby has been delivered at such a premature stage..*

We have not as of yet directly discussed the device with prospective parents or parents of extreme premature infants as we are in a preliminary phase of development and not in a position to offer the therapy. We have been gratified that in a few presentations to lay people and multiple presentations to neonatologists and maternal fetal medicine groups, there have been few objections raised and in general, the response has been overwhelmingly supportive. The points raised by this reviewer however are important and have occurred to us. We are designing the clinical device with those concerns in mind. The clinical device does not look like a bag, but rather very much like a standard nursery incubator. While the parent cannot touch the fetus, the unit will have 3D ultrasound capability, as well as a real time darkfield camera (because light will be very limited in the device) to allow the parents to visualize the fetus on a continuous basis if they wish. In addition, we plan to play maternal abdominal and heart sounds, and allow the parents to speak to the fetus etc. within the device if they wish so there will at least be some connection. We have added the point in the discussion as follows:

“An additional disadvantage that has been raised is parental perception of having their fetus in a “bag”. It is important to consider that the comparator is the extreme premature infant on a ventilator and in an incubator. We feel that parents will be relatively reassured that their fetus is being maintained in a relatively protective and physiologic environment. The clinical device will be designed with many features that should allow the parent to be connected with the fetus including ultrasound, a darkfield camera allowing real time visualization of the fetus within it's darkened environment, and the ability to play maternal heart and abdominal sounds to the fetus.”

- 2) *Glucocorticoids are usually administered to women with threatened preterm labour if delivery can be delayed >24h later. Their purpose of course is to mature the baby's lungs and other organs in preparation for extrauterine life. Given that most extremely preterm babies will have been exposed to maternal glucocorticoids, how will this affect an infant's ability to be EXITED and placed inside a biobag? Given over 40 years of glucocorticoid therapy to prevent RDS in preterm infants the concept of not using glucocorticoid therapy would be difficult to “sell” to clinicians..*

We have considered this and plan to administer glucocorticoids prior to placement on the device in the sheep system prior to clinical application to determine whether there are any unforeseen problems with doing so. Our feeling is that there is minimal potential harm in giving the glucocorticoids to patients prior to delivering them onto the device, so there is no reason for them to be withheld. We have added the following comment to the discussion to address this point and the next point:

“An important point is that placement on the system would not be prohibitive to standard

therapies for extreme premature infants. For instance, prenatal glucocorticoids could still be administered, and rapid delivery from the system and conversion to standard premature neonatal care would be necessary and warranted if the system failed.

- 3) *Many extremely preterm infants have been exposed to intrauterine infection which needs treatment. However, there are often long term consequences of that infection and the neurological damage that ensues is of particular concern. Are there interventions that would be provided to such infants that would not be possible if they were inside a biobag?*

As the primary intervention is antibiotics we do not feel that the system would be prohibitive to treatment of the fetus with chorioamnionitis. There are many questions that need to be answered however regarding our ability to clear contamination or treat infection within the system. We have been able to clear low grade contamination with antibiotics and increasing the rate of fluid exchange. We are developing an appropriate sheep model of chorioamnionitis to explore whether we can treat established infection in the system. We referred to this question somewhat in the previous version but have modified it slightly in the discussion to directly refer to chorioamnionitis.

“The ability to place an infant on the system after vaginal delivery, or to clear contamination under that circumstance, or circumstances of chorioamnionitis, remain to be experimentally determined, but would remain a possibility if umbilical vascular spasm can be prevented or reversed and contamination can be contained and/or fetal infection treated.

- 4) *Amniotic fluid is composed of a lot more than the relatively simple synthetic amniotic fluid that has been used here. It is a cocktail of hormones and growth factors as well as salts and nutrients derived from maternal tissues, the amnion and fetal urine. These contribute to fetal lung and gut development. Have the authors considered including some of these other factors?*

This is another excellent point raised by this reviewer that we have thought about a great deal and are investigating. We are developing a rabbit screening model for amniotic fluid exchange and hope to use this less expensive and time consuming model to test various components of amniotic fluid and their effect on fetal organ maturation for instance or nutritional effect, or resistance to infection. We have added the following statement to the discussion:

“Although we used a simple electrolyte solution in this study, amniotic fluid contains many trophic factors and other components that may be beneficial to the fetus. The development of an optimal “amniotic fluid” for use in the device is an important focus of our current research.”

Reviewer #5

We thank this reviewer for the favorable comments.

Reviewer #6

- 1) The authors have responded very favorably to almost all critiques of their original submission. There are two minor points which were not raised with the original submission that the authors should consider. One is that the procedure for placement of the UV catheter includes the final location of the tip of the catheters. Similar information for the UA catheters is not given in the text, and a brief mention of this in the text would be helpful.

We thank the reviewer for pointing out this deficiency. We have tried to clarify the position of the tip of the cannulas in the umbilical arteries and vein in the results section as follows:

“We developed a technique for umbilical cord vessel cannulation that maintains a length of native umbilical cord (5 to 10 cm) between the cannula tips and the abdominal wall, to minimize decannulation events and the risk of mechanical obstruction (Figure 1b and 1c). This interface is unique in that the umbilical and venous cannulas are only 2 cm long, most of which is used for securing the cannulas, therefore, the interface is functionally end to end.

2) Second, the authors correctly offer persistence of a fetal circulatory system as a major advance of their method over others. Although the authors describe the use of PGE and documentation of maintenance of the ductus arteriosus, they do not comment on assessment of ductus venosus patency.

We apologize for not making this clearer. In the previous version of the manuscript there is a color Doppler video (Video 3) that demonstrates a patent ductus venosus and this was observed throughout the runs. We have now added to the text in the results the following:

Daily echocardiography confirmed physiologic cardiac outputs and maintenance of the fetal cardiac circulation throughout the UA/UV runs, with near-normal ductus arteriosus flows (Figures 4a-c), patency and flow through the ductus venosus, and right to left shunting through the foramen ovale (Supplementary Information– Videos 2-4)²⁰.

Once again, we appreciate all three of the new referees willingness to review our revised manuscript and their expeditious review.